

# Modeling  Slope Environmental Lapse Rate (SELR) of temperature in the monsoon glacio-hydrological regime of the Himalaya

Renoj. J.Thayyen[1] and Ashok. P. Dimri[2]

[1]National Institute of Hydrology, Roorkee, Uttarakhand, India
[2]School of Environmental Sciences, Jawaharlal Nehru University, New Delhi, India

*Correspondence to: R. J. Thayyen ([renojthayyen@gmail.com](renojthayyen@gmail.com))*

**Abstract:** Moisture, temperature and precipitation interplay forced  through the orographic processes sustain and regulate the Himalayan cryospheric system. However, factors influencing the Slope Environmental Lapse Rate (SELR) of temperature along the Himalayan mountain slopes and an appropriate modeling solution remains a key knowledge gap. Present study evaulates the SELR variations in the monsoon regime of the Himalaya and proposes a modeling solution for
the valley scale SELR assessment. SELR of selected station pairs in the Sutlej and Beas basins ranging between 662 m a.s.l. to 3130 m a.s.l. and that  of  Garhwal Himalaya ranging between 2540 m a.s.l. and  3763 m a.s.l. were assessed in this study. Study suggests moisture- temperature interplay is forcing the seasonal as well as elevation depended variability of SELR. SELR constrianed to the nival- glacier regime is found to be comparable with the saturated adiabatic lapse rate (SALR)  and lower than the valley scale SELR.  Moisture influx to the region, during Indian summer monsoon (ISM) is
found to be lowering the seasonal valley scale SELR to SALR levels during July and August months. Highest valley scale SELR was observed in the months of April, May and June, which suseqently lowered to the SALR level with the influx of monsoon moisture. This seasonal variability of SELR is found to be closly linked with the variations in the local lifting condensation levels (LCL). Inter-annual variations in SELR of the nival- glacier regime is found to be significant while that of the  valley scale SELR is more stable.  Hence,  it is proposed to use the valley scale SELR for glacier melt/runoff
studies. We propose  a simple model for deriving the valley scale SELR of monsoon regime using a derivative of the Clausius–Clapeyron relationship. SELR modeling solution is achieved by deriving separate monthly SELR indices from one of the station pairs in the Beas basin from 1986-1990 and sucessfully applied for other select station pairs in Sutlej and Garhwal basins as well as for  different time periods. This work emphasis that the arbitrary use of temperature lapse rate is extremely untenable in the Himalayan region and significant further research is required to build data and concepts for a
comphrehensive atmospheric model valid across the glacio-hydrologic regimes of the Himalaya.

## 1 Introduction

The Hindu-Kush-Himalayan (HKH) mountain ranges play a very important role in regulating the climate and hydrology of the South - Asian region (Dey and Bhanu Kumar, 1983; Kumar et al.,1999; Zhao and Moore, 2004; Ye and Bao, 2005).



Sustenance of the large population in the region depends on the health of the rivers fed by this mighty mountain chain (Cruz et al., 2007; Bookhagen and Burbank, 2010; Immerzeel et al., 2010; Bloch et al., 2012). Acknowledgement of these facts has resulted in an increased focus on the Himalayan cryospheric systems, their response to changing climate and the ensuing impact on downstream flow regimes in recent years (Bookhagen and Burbank, 2010; Immerzeel et al., 2010; Kaser et al., 2010; Thayyen and Gergan,2010; Immerzeel et al., 2013). Empirical evidences of climate change over the Himalaya region are being presented and debated under various contexts. The IPCC (2007) report indicated a higher rate of glacier melting in the Himalayan region than elsewhere. However, systematic studies during the recent past have shown that the glacier change in the Himalayan region is comparable with other mountain glacier systems of the world except that of Karakorum region (Zemp et al., 2009, Bolch et al., 2012). Reported evidence of glacier expansion and slight mass gains in the Karakoram region during late 1990's and early twenty first century (Hewit,2005; Gardelle et al., 2012; Gardelle et al., 2013), decade long slightly positive or near- zero mass balance regime of the upper Chenab glaciers during 1990s (Azam et al., 2012; Vincent et al.,2013) and contrasting patterns of glacier mass balance change in the Himalayan regions during early twenty first century (Kaab et al., 2012) brings in more uncertainity about the processes driving the climate variability across the Himalayan arc. Other manifestations of climate change such as increase in temperature and decrease in precipitation are also evident in the region (Bhutiyani et al., 2007; Bhutiyani et al., 2010; Shrestha et al., 1999; Dimri and Dash, 2012; Shekhar et al., 2010; Duan et al., 2006). One of the key areas of knowledge gap over the Himalayan region is the moisture – temperature interplay at its higher elevations. While latitudinal control on the insolation sustains the polar cryospheric systems, the Himalayan cryospheric system is formed and sustained mainly by its high elevation as well as orographic processes. Hence, over the Himalayan region, insolation controls could be regulated by the regional physical-dynamical-thermodynamical processes associated with the mountain orography. Therefore, global climate change indicators could get modified through orographic processes over the Himalayan slopes and cryospheric systems; making it difficult to establish direct linkages between the two (Thayyen, 2013a). As mountain climate is a balance between free air advective processes and surface radiative effects (Whiteman et al.,2004; Pepin and Lundquist, 2008), unravelling the complex nuances of orographic controls on the Himalayan climate system is central to this understanding.

Many aspectes of elevation dependencies of surface temperature variations along the mountain slopes have been investigated across various mountain ranges of the world (Richner and Phillips, 1984; Rolland, 2003; Pepin and Seidel, 2005; Blandford et al.,2008; Kattel et al., 2013). Comparative studies of free air and surface temperature variations have amply demonstrated the significant differences between the two (Pepin and Losleben, 2002; Pepin and Seidel, 2005). Many studies highlighted the significant deviations of near surface temperature lapse rate of mountain slopes from the environmental lapse rate of 6.5K/km (Rolland, 2003; Marshall et al., 2007; Minder et al., 2010; Kirchner et al., 2013). However, while studying the larger tract of the ungauged high altitude Himalayan cryospheric regions, temperature lapse rates between ~6.0 to ~8.9 K/km are still being used arbitrarily to determine the higher altitude temperature values for snow/glacier melt modelling studies (Singh and Bengtsson, 2004; Rees and Collins, 2006; Kaser et al., 2010; Alford, 2010; Immerzeel et al., 2010; Immerzeel et al., 2013). Thayyen et al., (2005) showed a decrease in temperature lapse rate during peak monsoon months in the monsoon



regime and suggested that it could be driven by the latent heat released from monsoonal clouds. They cautioned the use of standard environmental lapse rate for snow and glacier melt studies in the high altitude regions dominated by monsoon systems, where peak melt period coincides with the peak of monsoon season. Later, Kattel et al. (2013) substantiated this process with regional scale assessment over the monsoon dominated regions of the Nepal Himalaya. Moreover, they observed a similar response during winter months as well. Earlier, Legates and Willmott (1990), Brazel and Marcus (1991) and De Scally (1997) also looked into the variations in the surface temperature lapse rate along the Himalayan slopes, which suggested a range of lapse rates extending from 10.8 K/km to 3.0 K/km.

Lack of understanding of the factors controlling the temperature variabiltiy over the mountain slopes leads to uncertainity over the warming rates of the mountainous region vis-a-vis the rest of the land surface (Rangwala and Miller, 2012; Beniston, 1997). Moreover, understanding the physical processes that control the temperature of the Himalayan slopes in different glacio-hydrological regimes (Thayyen and Gergan, 2010) is paramount to the understanding of the climate forcing on the Himalayan cryosphere and regional variability of emerging water scenarios. This understanding is also inevitable for climate downscaling over the higher Himalayan region for a better estimate of future climate trends. Modeling  near surface temperature lapse rate  is mainly achieved through  regression models (Bolstad et al., 1998; Rolland, 2003; Kattel et al., 2013). Yang  et al., (2011) attempted to integrate various factors influencing the  surface temperature lapse rate into a topoclimatic model. Regression equations were also used for downscaling large-scale weather parameters (Lundquist and Cayan, 2007; Gardner et al., 2009). However, factors controlling near-surface temperature lapse rate in various glacio-hydrological regimes of the Himalaya are largely unknown.

Presence and/or absence of moisture is key to the distribution of temperature and precipitation in an orographic system driving the climate of mountain slopes (Dimri and Niyogi, 2013)**.** The central and eastern Himalaya are impounded by moisture through Indian summer monsoon (ISM) during summer months (June – September: JJAS) (Kumar, 1999, 2006) and western and central Himalaya by Indian winter monsoon (IWM) during the winter months (Nov – Mar: NDJFM) (Dimri, 2013a, b). As these two systems negotiate the Himalayan region from opposite directions, the topography regulates their flow and produces seasonal moisture surplus and deficient zones across the Himalayan arc, forming distinct climatic and hydrological zones. These climatic and hydrological zones of the Himalaya are broadly classified into three units: (1) Himalayan system with dominant ISM, (2) Alpine system with dominant IWM and (3) Cold-arid system characterised by the absence of ISM in summer and subdued influence of IWM in winter (Thayyen and Gergan, 2010). In this paper we analyse the role of orography-moisture interplay in controlling the temperature distribution along the Himalayan slopes and high elevation cryospheric regions under monsoon regime and propose a modeling solution for estimating the SELR. The study area covers three basins such as Sutlej, Beas and upper Ganga. A brief description of general climate of these basins  is outlined in section 2. Data and methods are presented in section 3 and observational  results of  temperature, SELR and moisture  variations are presented  in section 4. A comparison of free environment lapse rate derived from ERA-interim, is also presented in this section 4. This is followed by the discussion on SELR modeling results in section 4.6, discussion on various processes governing the SELR in section 5 and conclusions are presented in section 6. The understanding developed





in this study could improve the efficiency and efficacy of climate and hydrological  models  in monsoon regime by better representation of the climate along the Himalayan slopes and cryospheric systems.

## 2 Study area and climate

Among the three dominant glacio-hydrologic regimes of the Himalaya, the present study focuses on the part of the monsoon regime of the western Himalaya coveing Beas, Sutlej and the upper Ganga basins (Fig. 1). Beas and Indian part of the Sutlej are nearby basins extending from $30^0$ 55' to $33^0$ 0' N and $76^0$ 0' to $79^0$ 0' E (Fig. 2a).  The Dingad catchment in the upper Ganga basin is located in Garhwal Himalaya, extending from $30^0$ 48' to $30^0$ 53' N and $78^0$ 39' to $78^0$ 51' E (Fig. 2b). These three basins have similar climatic charaterisitcs typical of the "Himalayan catchment", where climate is dominated by the ISM in summer  and IWM embedding western disturbances (WDs) in winter (Kumar et al., 1999, 2006; Dimri, 2009; Thayyen and Gergan,2010). Climate of the  upper regions  of Sutlej basin, draining the Spiti sub-basin, is significantly different from the lower reaches. Therefore, the present study is focusing on the monsoon dominted area of the basin where June to October precipitation constitutes around 62 to 80% of the annual precipitation. Mean annual precipitaiton recorded at select stations in the Beas-Sutlej basins range from 1224 mm at Kasol (662 m a.s.l.), 1950 mm at Manali (2050 m a.s.l.) and 330 mm at Kalpa (2439 m a.s.l.).    Stations in the Dingad catchment, Garhwal Himalaya experience comparable mean annual preciptiation ranging from 1485 mm at 2450 m a.s.l. and 1443 mm  at 3763 m a.s.l. Monthly precipitation and temperature distribution of these stations are shown in Fig. 3. All stations except Kalpa experience heavy precipitation greater than 300 mm during key monsoon months of July and August. Lower elevations (662 m a.s.l.) of the Beas-Sutlej basin experienced highest mean monthly temperature in the month of June (30.1ºC) followed by May (29.4ºC). However, higher elevation region experienced highest mean monthly temperature in the Month of July in the range of 25.4°C at 1089 m a.s.l. (Bhuntar), 21.8ºC at 2050 m a.s.l. (Manali), 18.4°C at 2439 m a.s.l. (Kalpa) and 15.5ºC at 3130 m a.s.l. (Rakcham). Second warmest month of these higher elevation stations is August. In the upper Ganga basin (Dingad catchment), highest mean monthly temperature recorded at different elevations during the study period was 18.6ºC in June  at 2540 m a.s.l. followed by 13.4ºC in July at 3483 m a.s.l. and 11.4ºC in July at 3763 m a.s.l. All these high values were recorded during the El-Nino year of 1998. During normal years, highest monthly temperatures of 17.4ºC, 12.4ºC and 11.3ºC were recorded at 2540 m, 3483 m and 3763 m a.s.l respectively.

## 3 Methodology

In the present work, we analyse the elevation control on temperature using data from three stations in the Sutlej basin, two stations in the Beas basin and  three stations in the Dingad catchment in the Garhwal Himalaya. Mean daily temperature data of Beas and Sutlej basins is  sourced from Bhakra Beas Management Board (BBMB). Data of Kasol (662 m a.s.l.),  Bhuntar (1089 m a.s.l.), Manali (2050 m a.s.l.), Kalpa (2439 m a.s.l.) and Rakcham (3130 m a.s.l.) were used in the study.  Kasol, Kalpa and Rakcham data of 1986-2007 and Manali-Bhuntar data of 1986-2000 were used to study the temporal and inter-annual changes in the temperature lapse rate. In the Dingad catchment, data collection were carried out under two separate



sponsored projects under the Indian Himalayan Glaciology Programme of the Department of Science and Technology (DST), Govt. of India during 1998-2004 period. Data of three stations located at Tela (2540 m a.s.l.), Gujjarhut (3483 m a.s.l.) and Basecamp (3763 m a.s.l.) (Fig. 2b) were utilised in the study. The Dingad catchment was monitored under Dokriani glacier project using standard India Meteorological Department (IMD) equipment. These were manual stations operated to cater to the multidisciplinary- multi institutional research programme. The data measured in these stations include Ambient Air Temperature, Wet Bulb Temperature, Wind Speed and Direction and Rainfall.

Deriving reliable precipitation information over the high altitude regions of the Himalaya, especially measuring snowfall is a big challenge (Menegoz et al., 2013; Shrestha et al., 2014). Hence, precipitation data is used only for describing the regional climatology. Solid precipitation collected in the standard rain gauges were measured as water equivalent after melting as per the India Meteorological Department (IMD) standard procedure. Further, standing snow depth and density were monitored four times during the December–April period at different altitudes along the valley bottom from Gujjar Hut to the Basecamp and accumulated snow water equivalent was calculated (Thayyen and Gergan, 2010). Many of these surveys conducted immediately after the snowfall events have shown that the precipitation measured by the rain gauge underestimate it by 26-32%. A study conducted in Nepal using tipping bucket rain gauge and differential change in the snow depth showed around 60% under reporting by the rain gauge (Shrestha et al. 2012). In this study, precipitation measurements were carried out at 0830 and 1730 hrs by manually melting the snow, which ensured no blockage at the mouth of the rain gauge during the snowfall periods. Here, monthly total precipitation values were arrived at by adding 30% to the precipitation measured by rain gauge during the peak snowfall season.

Air temperature data at Dingad stations were collected during the summer ablation months (May - October) of 1998-2004 (with an exception in 2002). For deciphering temperature lapse rate characteristics of the winter months, data collected during 2002-2004 period was used. Daily maximum and minimum temperatures were measured at 2540 m a.s.l. and 3763 m a.s.l. stations. However, significant data gaps exist in these data due to occasional instrument malfunction. Therefore, mean daily temperatures were derived for these stations by averaging the dry bulb temperatures measured at 0530, 0830, 1130, 1430, 1730 and 2030 IST. For a short period, mean daily temperature was calculated using the hourly temperature record from the thermograph and compared with the mean daily temperature calculated from dry bulb temperatures. On an average, mean daily temperature derived from dry bulb temperature is 1.0°C higher than the mean daily values derived from hourly temperature at Basecamp and 0.5°C higher at Tela station. For Dingad catchment, lapse rate has been calculated from these daily mean temperatures to maintain the consistency of the data throughout the study period.

All thermometers used in the manual measurement stations were factory calibrated as per the India Meteorological Department (IMD) standard and the final data were prepared by applying correction factors to the raw data. During the period of manual measurements, mean daily humidity was calculated from mean daily saturated and actual vapor pressures data were measured from dry bulb and wet bulb temperatures. Mean daily specific humidity was estimated by deriving mean daily actual vapour pressure from mean daily humidity (See annexure for detailed methods). Temperature measurements were carried out using calibrated thermometers and identical passive ventilation shields were used for stations in each region.



Hence, relative observational errors between the stations in each region are expected to be minimal. In addition, SELR is calculated as a difference of temperature between two elevation points and hence influence of marginal measurement errors gets minimized further in this analysis.

The average temperature decrease with height in the free atmosphere is generally called 'environmental lapse rate' (Berry, 2008), which usually approximates between 6 - 6.5 K/km. The temperature lapse rate along the mountain slopes significantly differs from the free atmosphere or environmental lapse rate (Marshall et al., 2007; Minder et al., 2010, Heynen et al., 2016). This temperature lapse rate along a mountain slope is termed here as the Slope Environmental Lapse Rate (SELR). The SELR variation is driven by naturally occurring orographic factors whereas environmental lapse rate of the free atmosphere is driven by vertical displacement of the air parcel. SELR between a pair of stations is calculated by the equation,

$$\text{SELR (K/km)} = \left( \frac{T_1 - T_2}{H_2 - H_1} \right)$$

Where $T_1$ and $T_2$ are the mean monthly temperature of lower and upper stations respectively in Kelvin, and $H_1$ and $H_2$ are the elevations of the lower and upper stations in kilometer. SELR is calculated for different altitude sections in the Sutlej, Beas and Dingad catchments to understand elevation dependency of slope lapse rate values of temperature vis-a-vis the moisture regimes. The lower and upper stations (Tela, 2540 m a.s.l. and Base camp, 3763 m a.s.l.) in the Dingad catchment are paired to deduce the SELR across the valley and termed as Section-1M. Higher elevation stations in the catchment are paired (Gujjarhut, 3483 m and Base camp, 3763 m a.s.l.) to derive the SELR of nival - glacier regime and termed as Section-2M. Similarly, in the Sutlej basins lower (Kasol, 662 m a.s.l.) and higher (Rakcham, 3130 m a.s.l.) stations are paired to calculate the valley scale (Section-1M) SELR and Kalpa, (2439 m a.s.l.) and Rakcham, (3130 m a.s.l.) stations are paired for representing the nival - glacier regime (Section-2M). In the Beas basin, section-1M is defined by paring Kasol (662 m a.s.l.) and Manali (2050 m a.s.l.) stations and Bhuntar (1089 m a.s.l.) and Manali (2050 m a.s.l.) stations constitute section-2M. In the following discussions, these sections will be referred as defined above. Present study is based on the mean monthly SELR derived from mean monthly temperature of the stations described above.

Liquid Condensation Level (LCL) is calculated to explore its relationship with the Slope Environmental Lapse Rate (SELR) using the following relationship

LCL= $(T_o - T_{do})/ (9.8 - T_d^2/158T)$

Where $T_o$ and $T_{do}$ are temperature and dew point temperature at the surface and $T$ and $T_d$ are temperature change with the altitude (Salby, 1996). Here, in the absence of higher altitude temperature values of the free atmosphere, surface values are used with little error.



### 3.1 Modeling SELR for monsoon regime

The environmental lapse rate of 6.5K/km is the linear lapse rate of the troposphere considered as one single layer in the U.S standard atmosphere (NOAA, 1976). The equations governing the DALR and SALR are well established. However, these equations are generally used in the context of air parcels lifted 'vertically' upwards under different moisture conditions. Though various studies pertaining to lapse rate over other mountainous region prevail (Thyer, 1985; Rolland, 2003; Harlow et al., 2004; Blanford et al., 2008; Minder et al., 2010) a model which encapsulates the mountain process remains a challenge. Over central Himalaya, Kattel et al. (2013) has provided lapse rate estimation with functional reference of the elevation. Here, we are proposing a simple solution to capture the seasonal SELR variations observed in the monsoon regime with a hypothesis that the moisture influx during the monsoons and lack of it during rest of the period, especially during April, May & June (AMJ) play a decisive role in determining the valley scale SELR and thereby the temperature distribution along the valley slopes.

The DALR and SALR of the free atmosphere is governed by the following equations respectively (Robinson and Henderson-Sellers, 1992),

$$\frac{dt}{dz} = -\frac{g}{C_p} \tag{1}$$

$$\frac{dt}{dz} = -\frac{g}{C_p} - \left[\left(\frac{L}{C_p}\right) x \left(\frac{dw_s}{dz}\right)\right] \tag{2}$$

where, dt/dz is rate of change of temperature (T, Kelvin) with height (z), g is acceleration due to gravity (9.8 m/s²), $C_p$ is specific heat at constant pressure (1.004 J/g-K), L is the latent heat of phase change (L= 3071 − 2.134 T J/g), $w_s$ is the saturation mixing ratio.

$$w_s = 0.622 \left(\frac{e_s}{p}\right) \tag{3}$$

Where $e_s$ is saturation vapor pressure and $p$ is the mean station pressure in hPa.

To solve this equation, temperature data at two elevations are required for estimating the change in the saturation mixing ratio. Even though this equation provides valuable insight of processes governing the temperature lapse rate under saturated conditions, this equation does not have the predictive advantage. What is required for a snow/glacier model is a SELR model, which delivers the higher elevation temperature distribution based on a single base station temperature data at the lower elevation. Following manifestation of the Clausius–Clapeyron relationship is found to be appropriate for this purpose (Peixoto and Oort, 1992).

$$\frac{dT}{dz} = -\frac{g}{C_p} \left[\frac{(P+N)}{\{P+(\varepsilon L / CpT)N\}}\right] \tag{4}$$



where $N = \dfrac{\varepsilon L \mathrm{es}}{RT}$ (5)

where, T is the base station temperature in Kelvin, P is mean base station pressure in hPa, $\varepsilon$ is the ratio between molecular weight of the water and dry air (0.622) and R is gas constant for dry atmosphere (0.287J $g^{-1}$ $K^{-1}$). There are different forms of this equation available in the literature. We chose this equation due to its explicit reference to the elevation of the temperature measurement station. However, difference between modelled and observed SELR is still found to be significant and devoid of observed seasonal variations characterizing the monsoon regime.

To improve the response of the Eq. 4, the fraction of moisture potentially responsible for forcing the observed SELR (*dwsf*) has been calculated using the observed SELR and estimated SALR using Eq. 2 as described below:

$$\frac{dwsf}{dz} = \frac{\left[\dfrac{dT}{dz}\right]_{obs} - \left[\dfrac{dT}{dZ}\right]_{equ}}{\dfrac{L}{C_p}}$$ (6)

Where, *dwsf* represents the potential withdrawal/influx of moisture from/to the elevation section with reference to the respective SALR.

By using this information, we have derived monthly SELR indices (*Mi*) for section-1M of the monsoon region as follows:

$$Mi = \frac{\sum\limits_{n=1}^{n} dwsf}{\sum\limits_{n=1}^{n} dws}$$ (7)

These monthly indices are applied to the Eq. 5 and modified 'N' has been calculated as shown in the Eq. 8 below. Eventually this newly derived 'N' is applied to the Eq. 4 to derive the SELR for different sections under different basins of the monsoon regime.

$$N = \frac{\left[es - (esMi)\right]\varepsilon L}{RT}$$ (8)

Monthly SELR indices (Table 3) calculated for the section 1M (Kasol-Manali) for a period of 1986-1990 is used for further testing of the SELR model for section-1M of all the three basins (Beas, Sutlej and Dingad) for different time periods. The 10 year mean monthly SELR of the Sutlej and Beas basin (1991-2000) and the mean monthly SELR for 5 individual years were tested and it was demonstrated that the indices derived at one section are suitable across the monsoon regimes under study. Model performance has been tested by calculating the p-value and Root Mean Square Error (RMSE).





## 4 Results

### 4.1 Valley scale SELR variations

Seasonal variations in the valley scale SELR (Section 1M) of all the three basins under monsoon regime (Sutlej, Beas and upper Ganga basins) have shown remarkable similarity. Lowest SELR was recorded during monsoon months of July and August and highest SELR was observed during April and May months. SELR of winter months was also significantly lower than that of April and May months in all the basins (Fig.4). Annual range of mean monthly SELR is significant in all the three basins extending from 2.5 to 9.2 K/km for Kasol-Manali section (Beas basin), 3.9 to 9.5 K/km for Kasol- Rakcham section (Sutlej basin) (Table 1) and 1.9 to 9.0 K/km for Tela – Base camp section(Upper Ganga basin) (Table 2). This clearly suggests that the practice of using standard environmental lapse rate of 6.5 K/km across the seasons is untenable for this region. Presence of monsoon during the peak glacier melt period of July and August is the charactersitcs of the 'Himalayan catchment'(Thayyen and Grgan, 2010). Therefore, lower SELR during these months in the region has significant impact on the melting of cryopsheric elements including glaciers as it facilitates a warmer climate over the higher elevation regions. Long-term mean SELR (1986-2000) of Kasol-Manali sections duirng monsoon months of July and August were 4.4 and 4.1 K/km respectively. For Kasol- Rakcham section it stands at 5.0 K/km for both the months and for Tela- base camp section, mean SELR of July and August months were 5.3 and 5.4 respectively. Similar behavior of SELR is reported from a large number of pair of stations in Nepal Himalaya as well (Kattel et al., 2013). Hence, we strongly believe that this is a regional phenomenon characterizing the influence of monsoon moisture influx to the region and resultant orographic lift to the higher elevations in the 'Himalayan catchments'. Monsoon transition month of September continues to experience predominantly lower lapse rate and post monsoon month of October experiences predominantly higher (>5.8° K/km) SELR. We also looked into the inter-annual variability of valley scale SELR of Kasol-Manali and Kasol-Rakcham sections using 15-20 years of available data. The coefficient of variation of SELR for valley scale SELR ranges from 0.06 to 0.17, which suggests a high inter-annual stability of valley scale SELR (Fig.5).

### 4.2 SELR variations in the nival-glacier regime

SELR of section -2M representing the nival- glacier regime (Fig.2) is considered important for cryospheric system studies in the Himalaya. Generally, studies on Himalayan glaciers focus on the data collection of glacier regime following the benchmark glacier monitoring strategy (Fountain et al., 1997). It is well known that the monsoon precipitation declines as we move up the higher elevations (Bookhagen and Burbank, 2010). Characteristics of the SELR response constrained within these elevation zones acquire significance under these circumstances. It is can be seen that the SELR of nival-glacier regime (section -2M) of all the three basins is significantly different from the SELR of valley scale SELR. Section 2M is characterized by lower SELR across the seasons (Fig.4). Absence of significant intra-seasonal SELR variation is also noted as a key difference from Section-1M. Due to this, 'monsoon lowering' of valley scale SELR is absent from section -2M of Beas and Dingad basins. However, section- 2M of the Sutlej basin does experience subdued monsoon lowering. Long- term





(1986-2000) monthly mean SELR of section-2M of Beas basin (Bhuntar-Manali) range between 4.4 -2.5 K/km and that of Kalpa -Rakcham (1986-07) vary between 5.5 to 3.4 K/km. For Dingad catchment, section-2M data is available only for summer months (MJJASO) and ranged between 2.1 and 4.6 K/km with few exceptions. Major exception was in 1998, when SELR of section-2M was significantly higher than other years (3.6 to 7.1 K/km) during the ablation/monsoon months (June - September). As mentioned earlier, 1998 was an El-Nino year and there could be large scale teleconnections influencing such behavior. However, those aspects are not investigated in the present study. This station pair also reported temperature inversion almost every year in the month of November.

One major observation regarding the SELR of nival- glacier regime is its weak inter-annual stability. Coefficient of variation of section - 2M SELR is found to be greater than section-1M (Fig.5) throughout the year and ranges between 0.46 and 0.16. With lack of supporting data from these stations, possible factors driving these changes could not be ascertained. This result highlights the need for significant further research to build data and concepts for a comphrehensive atmospheric model valid across the Himalaya. Irrespective of the forcing, the higher inter-annual variation of SELR of the nival-glacier regime suggests its limited utility in modeling the hydrological response of the system, especially in the case of modeling the future scenarios of streamflow and glacier change.

### 4.3 SELR and specific / relative humidity relationship

Amount of water vapor in the atmosphere and its seasonal variations could be playing an important role in forcing the SELR variations. Some insights on these aspects are developed from the data generated from Dingad catchment. Higher specific humidity during monsoon months of July and August is characteristic of this regime. Mean monthly specific humidity at 2540 m a.s.l. varies between 3.59 to 16.4 g/kg during the observation period. Highest observed value of specific humidity was 18.17 g/kg in the month of July 1998. At higher elevation (3763 m a.s.l.) mean monthly specific humidity ranged between 2.92 g/kg to 11.4 g/kg. It is noted that the water vapor in the atmosphere during winter months (NDJFM) ranges from 2.92 to 5.3g/kg and is significantly lower (Fig.6). It is observed that the higher elevation stations generally experience lower specific humidity throughout the year. Feld et al., 2013 also observed a similar response in Sierra Nevada Mountains in California. Monthly mean relative humidity of the uppermost station in the monsoon regime (3763m a.s.l.) remained above 80% during the peak monsoon months (Fig.6) leading to sustained lower SELR of section-2M during these months. Whereas relative humidity at the lower station (2540m a.s.l.) fluctuates between 65 to 80% during the summer months.

### 4.4 Comparison with free environment temperature lapse rate of the study areas

Environmental lapse rate of the free atmosphere is calculated from the ERA-interim reanalysis (Dee et al., 2011) (https://apps.ecmwf.int/auth/login/), and regional climate model (RCM) from the Max Planck Institute for Meteorology (REMO) (Jacob et al., 2007) for comparing with the observed SELR from station data. REMO model simulation uses global ERA-interim reanalysis data to supply large-scale boundary information. The RCM inputs will be important information for providing finer scale regional climate accounting for regional feedbacks, physical processes and dynamical forcing. Model



details are not provided here as it is not the core focus of this paper. Discussion is limited to the comparison with corresponding initial and boundary conditions from ERA-interim and corresponding station observations. So far no radiosonde ascents have been performed in the regions of this study. It is important to mention here that various reanalysis are amalgamations of observed station records, and satellite information, and uses different mathematical and statistical algorithms to generate the reanalysis data. These are also not discussed here in detail as it is out of the scope of the present work. However, ingenuity of the ERA-interim data over other reanalysis data is proven as it uses observed surface temperature records during its preparation (Simmons et al., 2004, 2010). This particular fact is very important for the Himalayan region due to paucity of observation network and may give a benchmark for future research in the absence of such records. Corresponding ERA-interim forcing to simulate REMO regional climate model is expected to provide more exhaustive information over the study region. However, it is for the first time that modeling has been employed after ERA-interim data records for the period 1998-2004 for monsoon regime at Tela (30º51'26.22" N, 78º40'39.96"E) and Sutlej/ Beas (31º15'- 33º0'N, 77º30'-79º0' E) for 2000-2007 period are extracted to compare with the corresponding station observation. During monsoon months observed and ERA-interim environmental lapse rate matches well with summer monsoon lowering for valley scale SELR for all the three basins (Fig. 7). For Dingad catchment November to April shows significant difference between the two; whereas for Sutlej/Beas basin, more difference is observed during April to June. It is observed that the ERA-Interim gives same lapse rate for different altitude sections and fails to capture the unique SELR response of the nival-glacier section (section-2M). Environmental lapse rate derived from REMO follows the same trend of ERA-interim but with higher values. Hence, REMO has not been tested for Sutlej/ Beas basin. Whiteman and Hoch (2014) showed that the relationship between the pseudo-vertical temperature and radiosondes improves with elevation and steepness of the slope. It is important to mention here that ERA-interim is at 1ºx1º lat/lon horizontal resolution, which is too coarse over the region of study with heterogeneous land use and variable topography. It could be inherent that during the preparation of the reanalysis, most of the sub-grid scale processes are not being captured within the resolution of reanalysis. However, it is obvious that environmental lapse rate based on ERA-interim and REMO are sensitive to moisture. It is evident that both ERA interim and REMO gives comparable temperature lapse rate during the monsoon months. Study suggests that the gridded reanalysis data captures some key regional processes such as monsoon lowering of valley scale SELR but fails to capture pre-monsoon response of higher SELR as well as lower SELR of nival-glacier regime. Fiddes and Gruber (2014) have extensively shown the downscaling method of climate variables from coarser to finer resolution over heterogeneous topographic regions.

## 4.5 Modeling valley scale SELR

Extrapolation of temperature from lower elevation to higher elevation is imperative for modeling glacier mass balance and snow/glacier melt runoff and variety of other studies where temperature is a critical factor. Assessment of future temperature variation along the high elevation nival-glacier system for climate change studies is also highly influenced by the ability to model the seasonal variability of SELR. Developing better insights on the factors controlling the SELR on temporal and





spatial scale is critical for understanding the climate forcing on different elevations of the Himalaya. Hence, SELR modeling attempted here is the first step in this direction where we tested the model performance in spatial and temporal scale within the selected monsoon basins of the Himalaya. SELR indices developed for one specified section of the Beas basin (662-2050 m a.s.l.) during 1986-1990 period is tested for Sutlej as well as the upper Ganga basin without any further calibration.

Performance of the model is also tested for different time period to assess its utility in studying the future climate change and its impact.

SELR modeling is attempted only for valley scale lapse rate (Section-1M) as lapse rate of nival-glacier system (Section-2M) is found to have higher inter-annual variability as discussed in section 4.2 above. Equations 2 and 4 were implemented to test its response along sections-1M of Beas basin (Kasol-Manali) using mean monthly temperature and SELR of 1986-90 period.

Seasonal variability is more pronounced for Eq. 2 as it uses the estimate of change in saturation mixing ratio between the two stations for calculating the heat generated due to condensation process. On the contrary, Eq. 4 is more stable as it uses saturation vapor pressure value of single station for calculating the potential heat generated during condensation (Fig.8). Both the models are in better agreement with the observed SELR in winter (DJ) and monsoon months (JA). April, May and June months experienced highest deviation between observed SELR and calculated SALR. It is noted that the observed

SELR values were higher than the calculated SALR values for the Section-1M. These deviations are resolved by developing SELR indices (Fig.8). Developed indices for the Kasol (662 m a.s.l) - Manali (2050 m a.s.l.) section for 1986-90 period is given in Table-3. Derived monthly indices were tested in the same section by using decadal mean monthly temperature of Kasol (662 m a.s.l.) for 1991-2000 period. The model performance is found to be excellent with a p-value of $5.81 \times 10^{-9}$ and an RMSE of 0.35 K/km with $r^2$, 0.98 (Fig.9). Further, the model is tested for five individual years from 1996-2000, which

also showed good performance of the model with p-value ranging from $1.77 \times 10^{-4}$ to $9.89 \times 10^{-3}$ and corresponding $r^2$ ranging from 0.86 to 0.70 and RMSE from 0.70 to 1.17 K/km (Table 4). To assess the regional validity of the derived indices, the model is tested for section-1M of the Sutlej basin (Kasol- Rakcham). Testing was undertaken with decadal mean monthly temperature and SELR (1991-2000) as well as for individual years from 2003 to 2007. Model has performed very well with p-value ranging from $1.0 \times 10^{-5}$ to $4.3 \times 10^{-3}$ with $r^2$ ranging from 0.92 to 0.74 and RMSE ranging from 0.43 to 0.85 K/km.

Further, the indices were tested for the upper Ganga basin for two full years (2002-03 and 2003-2004) and four years of summer ablation months (May-October). Results show good model performance with p-value of $1.09 \times 10^{-4}$ to $3.91 \times 10^{-3}$ and $r^2$ ranging from 0.95 to 0.75(Fig.9). RMSE values for this section are found to be higher and range from 1.09 to 1.23 K/km. Large difference between the altitude range for which the indices were developed (662-2050 m a.s.l.) and that of the Dingad catchment (2540-3763 m a.s.l.) needs to be taken into account. In general, the SELR indices derived for the

monsoon regime are useful for deriving the monthly/seasonal SELR of the region effectively.

## 5 Discussions

Comparable SELR variations observed in the Beas, Sutlej and upper Ganga basins and Nepal (Kattel et al., 2013) suggest that the processes controlling the SELR are similar across the monsoon regime. Long-term consistency in seasonal response



of SELR of monsoon regime as shown in the present study also underlines this aspect. Higher SELR of valley scale (Section-1M) than that of the nival- glacier regime (Section-2M) is one of the key response identified for the first time. Apart from this, higher valley scale SELR during April, May and June months and SELR lowering during key monsoon months of July and August are suggested as the characteristics SELR response of the monsoon regime. Seasonal variations in the influx

of moisture into the 'Himalayan catchments', its orographic lifting and resultant latent heat released during condensation could be guiding these responses. Significant proximity of SELR of Section-1M with that of the theoretical SALR for corresponding pressure levels during the Indian Summer Monsoon (ISM) months is indicative of this process (Fig.10). In winter months, monsoon regime receives moisture from the IWM. Combined with the prevailing low winter temperature, SELR shows closer values with that of corresponding SALR in all the three basins. This suggests that the observed seasonal

variations in the valley scale SELR are linked to the major meso-scale weather systems (IWM and ISM). Study by Kattel et al. (2013) showing similar results under monsoon climate from a study of 56 station data in Nepal also suggests SELR linkages with these weather systems and its wider spatial extent. Dominance of the large scale circulation over local slope and valley winds on temperature lapse rate is reported from Alps also (Kirchner et al., 2013).

Mean monthly SELR of nival-glacier regime (Section-2M) matches well with the plausible SALR for corresponding pressure levels throughout the year (Fig. 10). Winter lowering of SELR leading to lesser temperature difference between lower and higher elevation do not have significant influence on the regional hydrology or glacier characteristics because the ambient temperature in these region is well below the freezing point in winter and the snow and glacier regions remain under the non-melt regime. However, in summer, SELR lowering forced by the moisture influx and orographic up draft greatly

influence the melt processes of glaciers and snow cover. Higher inter-annual variability of nival- glacier regime SELR as observed for Beas and Sutlej basins indicate more complex processes driving these changes and raise questions regarding its use for modeling snow and glacier melt, especially for modeling future runoff and glacier fluctuations. Various researchers have shown that the glacier and snowmelt estimation by degree day method is highly sensitive to the near surface lapse rate (Gardner and Sharp, 2009; Marshall et al., 2007 and Immerzeel et al., 2014). This result points towards the need for re-

visiting the benchmark glacier monitoring strategy (Fountain et al, 1997) for mountain glaciers. Under this strategy, climate monitoring for glaciological studies are focused on the glacier regime, which limits our ability to understand/incorporate key orographic processes at lower elevations forcing the temperature variations at higher elevations. The results suggest that the use of valley scale SELR having higher inter-annual stability is more appropriate for extrapolating the temperature to the higher elevations as of now.

Higher SELR observed during the month of April, May and June is important for snow/glacier melt modeling. Snow melt is the dominant runoff component of mountain streams during these months and also has significant impact on the glacier mass balance. In case of displacement of air parcel along a vertical air column, such variations in the lapse rate occur above and below the lifting condensation level (LCL) (Ahrens, 1991). Analysis in the Sutlej and Dingad catchment suggests that the same processes are followed by the air parcel while being lifted along the mountain slopes by the orography as well.



Significant correlation between LCL at 2540 m a.s.l. and SELR of Section-1M ($r^2$, 0.59 to 0.72, $P <0.001$) in the monsoon regime during the observation years suggest that the seasonal LCL height variation plays a dominant role in determining the SELR, especially in the pre-monsoon and monsoon periods. In both the basins, LCL in summer/monsoon months is found to be closer to the land surface, forcing SELR towards SALR (Fig. 11). On the contrary, LCL shifts to the higher altitudes during moisture deficit months of April, May and June in the pre-monsoon period forcing higher SELR for Section-1M shifting towards the DALR in all the three basins under study (Fig.10). A major process consuming significant energy within the parcel is the re-evaporation of condensed precipitation while falling through the warmer layers below (Dolezel, 1944). We propose that the rate of re-evaporation of water droplet, governed by the seasonal variations in the LCL could be playing an important role in determining the seasonal variations in the valley scale (Section-1M) SELR. The net energy released through the condensation of summer monsoon moisture is suggested to be the prime driver of SELR along the Himalayan slopes during this season. Amount of water vapor in the atmosphere during winter months is significantly less than the summer months and higher elevations have less water vapor as compared to the lower elevation stations (Fig.6a). However, lower temperatures ensure condensation and higher sustained relative humidity at high altitudes. We suggest that the higher LCL and higher valley scale SELR during April, May and June months is very critical for higher elevation cryospheric region as it forces a shift in the warmest months from May– June at lower elevations to July– August in the higher elevations as pointed out earlier. This facilitates persistence of snow over the mountain slopes and glaciers for longer period during these post-winter months. SELR variations could also be influenced by factors such as land surface conditions (Pepin and Losleben, 2002; Pepin and Kidd, 2006). However, Kirchner et al. (2013) found no significance difference between snow and no snow cover days in lapse rate based on daily mean temperature. Snow and non-snow conditions shall produce most significant contrasting land surface conditions influencing the ambient temperature measured at screen heights in the mountains. In the Dingad catchment, differing snow cover conditions were experienced in the year 1998 and 1999 (Thayyen and Gergan, 2010). However, the valley scale lapse rates of all the measurement years, especially of these two years were similar. Hence, we believe that the variation in the valley scale SELR is governed by the orographic processes rather than surface conditions. Long-term stability of the observed valley scale SELR underlines this fact. However, change in the surface conditions may be playing a significant role in the SELR of nival-glacier regime as suggested by the higher inter-annual variability. These results point towards the need of more focused research on the moisture- temperature interplay in controlling the temperature changes in the Himalayan region. Development of a full atmospheric model to capture the SELR variability across the Himalayan altitudes, including the SELR variations in the nival-glacier regime is imperative for linking it with future evolution of nival- glacier regime of the Himalayas. Study suggests that the use of standard environmental lapse rate for extrapolating the temperature to the higher elevation regions of the Himalaya for snow and glacier melt calculations have no physical basis. Proposed modeling solution is a good beginning in this direction.



## 6 Conclusions

Significant seasonal variation of observed slope environmental lapse rates (SELR) in the monsoon regime of the Himalaya amply demonstrates that the use of standard environmental lapse rate for temperature extrapolation is not appropriate. This is more so in the nival –glacier regime, where SELR is consistently lower than the standard environmental lapse rate.

Seasonal SELR variations of the monsoon regime are found to be linked with the presence or absence of moisture and its condensation regime. Seasonal moisture influx during the winter and summer monsoon period forces lowering of the SELR. Manifestations of atmospheric pressure-moisture-temperature variability driven by the orographic lifting leads to greater saturation at the higher elevation regions. This results in comparatively lower SELR's in the higher elevations. Seasonal variation in the height of the lifting condensation level (LCL) is found to be influencing the seasonal variations of SELR.

Proposed model with monthly SELR indices has provided a simple process based solution for calculating SELR. Local surface energy balance including net radiation and turbulent heat fluxes are believed to be the primary determinant of surface temperature and its vertical gradient (Marshall et al., 2007). However, distinct surface temperature gradients along the mountain slopes observed for the wet systems of the Himalaya clearly indicate that the seasonal variations in the moisture availability and the condensation regimes have an overriding influence in determining the SELR and thereby the temperature

distribution in an orographically driven system. The valley scale lapse rate is found to be much stable inter-annually as compared to the nival-glacier system SELR and calls for a relook in the glacier monitoring strategy for the region. New insights presented in this study may help to improve our understanding of the climate-cryosphere interaction in the monsoon regime and improvement in the snow/glacier runoff modeling in the Himalayan basins, better understanding of the climate change impact on the Himalayan slopes and more realistic climate downscaling in the region. Present study indicates that the

global climate change and its manifestations could be impacting the higher Himalayan regions through the orographic modulations. Hence, developing robust understanding of future climate trajectory over the Himalaya requires better understanding of the moisture - temperature - orography interplay.

## Author contributions

RJT conceived the study, collected field data, conducted the analysis and prepared the manuscript. APD contributed in developing the SELR modeling concept and MS preparation.

## Acknowledgements

RJT acknowledges the support and encouragement of Shri. R.D Singh, Director, NIH, and Dr. S.K Jain, Head WRS Division

and Dr. Sanjay Jain, NIH. Authors acknowledge Ms Anila Romil for providing English edits of the manuscript. Financial support from Department of Science & Technology, Govt. of India under project No's. SR/DGH/PK-1/2009 & ES/91/23/97 is duly acknowledged.



**Annexure**

a)  Saturated vapor pressure from ambient temperature (T°C)

$\qquad E_{s} = 6.11 \times e^{(7.5 \times T)/(237.3 + T)}$

b)  Dew point temperature from wet bulb temperature/Relative humidity

$Td = (237.3 \times B)/ (1-B)$
$B = (\ln (E/6.11))/17.27$
$E = Ew - (0.00066 \times (1+ 0.00115 \times Tw) \times (T-Tw) \times P)$
$Ew = 6.11 \times e^{(7.5 \times Tw)/237.3 + Tw)}$

Ew = Saturation vapor pressure at wet bulb (hpa)
E = Actual vapor pressure (hpa)
Tw = Wet Bulb temperature (°C)
P = Station barometric pressure, hpa
B = intermediate value (no units)
Td = Dew point temperature (°C)

c)  Actual Vapor pressure (hpa) from dew point Temperature (Td)

$\qquad e = 6.11 \times 10^{(7.5 \times Td)/237.3 + Td)}$

d)  Mixing ratio

$W = 0.622e (P - e)^{-1}$
$Ws = 0.622e_{s} (P - e_{s})^{-1}$
W mixing ratio, Ws Saturated mixing ratio and  P ambient air pressure (hpa)

e)  Relative Humidity (RH%) = W/Ws x 100

f)  Specific Humidity (gm/kg)

$\qquad = 1000 \left[ \dfrac{(\epsilon e)}{\{P - ((1-\epsilon)e)\}} \right]$

g)  Latent heat of vaporisation    L= 3071 – 2.134 T J/g

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



**Table 1.** Mean valley scale temperature SELR (Section-1M) in the Sutlej and Beas basins

| Station pair | Kasol-Manali | Kasol-Rakcham | Bhuntar-Manali | Kalpa-Rakcham |
|---|---|---|---|---|
| Elevation range (m a.s.l.) | 662-2050 | 662-3130 | 1089-2050 | 2439-3130 |
| Period | 1986-2000 | 1986-07 | 1986-2000 | 1986-07 |
| Nov | 6.3 | 6.2 | 3.4 | 5.2 |
| Dec | 5.3 | 5.9 | 2.5 | 4.4 |
| Jan | 5.2 | 6.0 | 3.0 | 4.2 |
| Feb | 5.9 | 6.5 | 3.8 | 3.6 |
| Mar | 6.8 | 7.1 | 4.0 | 4.0 |
| Apr | 7.4 | 7.6 | 3.2 | 4.7 |
| May | 8.1 | 7.4 | 3.8 | 4.7 |
| Jun | 7.0 | 6.6 | 4.4 | 5.2 |
| Jul | 4.4 | 5.0 | 3.7 | 4.0 |
| Aug | 4.1 | 5.0 | 3.7 | 3.4 |
| Sep | 5.0 | 5.7 | 4.0 | 5.2 |
| Oct | 6.4 | 6.5 | 4.1 | 5.5 |



**Table 2**. Slope Environmental Lapse rate (SELR) of temperature in the monsoon regime between 2540 m and 3763 m a.s.l. (Section-1M) of the Dingad catchment, Upper Ganga basin

| | SELR(K/km): 2540-3763 m a.s.l. Section-1M | | | | | | |
|---|---|---|---|---|---|---|---|
| Months | 1997-98 | 1998-99 | 1999-2000 | 2000-01 | 2001-02 | 2002-03 | 2003-04 |
| November | 5.4 | ND | 6.6 | ND | 5.4 | 5.5 | 5.7 |
| December | ND | ND | ND | ND | 5.3 | 5.1 | 5.3 |
| January | ND | ND | ND | ND | 6.0 | 4.9 | 5.7 |
| February | ND | ND | ND | ND | 6.1 | 5.6 | ND |
| March | ND | ND | ND | ND | ND | 6.0 | 6.5 |
| April | ND | ND | ND | ND | ND | 5.9 | 6.2 |
| May | 7.1 | 6.7 | 6.0 | ND | ND | 6.5 | ND |
| June | 7.0 | 6.0 | 5.6 | 5.6 | ND | 6.4 | 6.3 |
| July | 5.8 | 5.2 | 5.1 | 5.0 | ND | 5.4 | 5.6 |
| August | 5.5 | 5.5 | 5.0 | 5.3 | ND | 5.0 | 5.3 |
| September | 6.1 | 5.6 | 5.6 | 6.0 | ND | 5.5 | 5.6 |
| October | 6.3 | 6.2 | 5.7 | 6.5 | ND | 6.0 | 6.0 |

ND : No Data





**Table 3**. Monthly SELR indices ($Mi$) derived from Kasol (662 m a.s.l.) – Manali (2050 m a.s.l.) section using monthly mean temperature data of 1986-1990 period for representing the monsoon regime of the Himalaya.

| SELR Indices (Kasol-Manali) (662-2050 m a.s.l.) | |
|---|---|
| Months | SELR-Indices |
| Nov. | 0.63 |
| Dec | 0.18 |
| Jan | -0.02 |
| Feb | 0.44 |
| March | 0.66 |
| April | 0.84 |
| May | 0.90 |
| June | 0.84 |
| July | 0.36 |
| August | 0.19 |
| Sept | 0.45 |
| Oct | 0.73 |



**Table 4**. Details of error analysis of the SELR model for Beas, Sutlej and Upper Ganga basins.

Kasol- Manali (662-2050 m a.s.l.)

| Year | $r^2$ | RMSE k/km | p-Value |
|---|---|---|---|
| 1991-2000 | 0.98 | 0.35 | 5.81E-09 |
| 1996 | 0.86 | 1.17 | 1.77E-04 |
| 1997 | 0.85 | 0.70 | 2.41E-04 |
| 1998 | 0.81 | 0.79 | 9.69E-04 |
| 1999 | 0.86 | 0.70 | 1.96E-04 |
| 2000 | 0.70 | 0.88 | 9.89E-03 |

Kasol- Rakcham (662-3130 m a.s.l.)

| Year | $r^2$ | RMSE k/km | p-Value |
|---|---|---|---|
| 1991-2000 | 0.89 | 0.43 | 5.7E-05 |
| 2003 | 0.90 | 0.72 | 3.6E-05 |
| 2004 | 0.74 | 0.74 | 4.3E-03 |
| 2005 | 0.75 | 0.85 | 3.7E-03 |
| 2006 | 0.92 | 0.48 | 1.0E-05 |
| 2007 | 0.83 | 0.82 | 5.3E-04 |

Tela-Base Camp (2540-3763 m a.s.l.)

| Year | $r^2$ | RMSE k/km | p-Value |
|---|---|---|---|
| 1998(May-Oct.) | 0.94 | 1.14 | 1.73E-03 |
| 1999(May-Oct.) | 0.94 | 1.23 | 1.49E-03 |
| 2000(May-Oct.) | 0.95 | 1.17 | 1.16E-03 |
| 2001(May-Oct.) | 0.92 | 1.18 | 3.15E-03 |
| 2002-03 | 0.87 | 1.12 | 1.09E-04 |
| 2003-04 | 0.75 | 1.09 | 3.91E-03 |



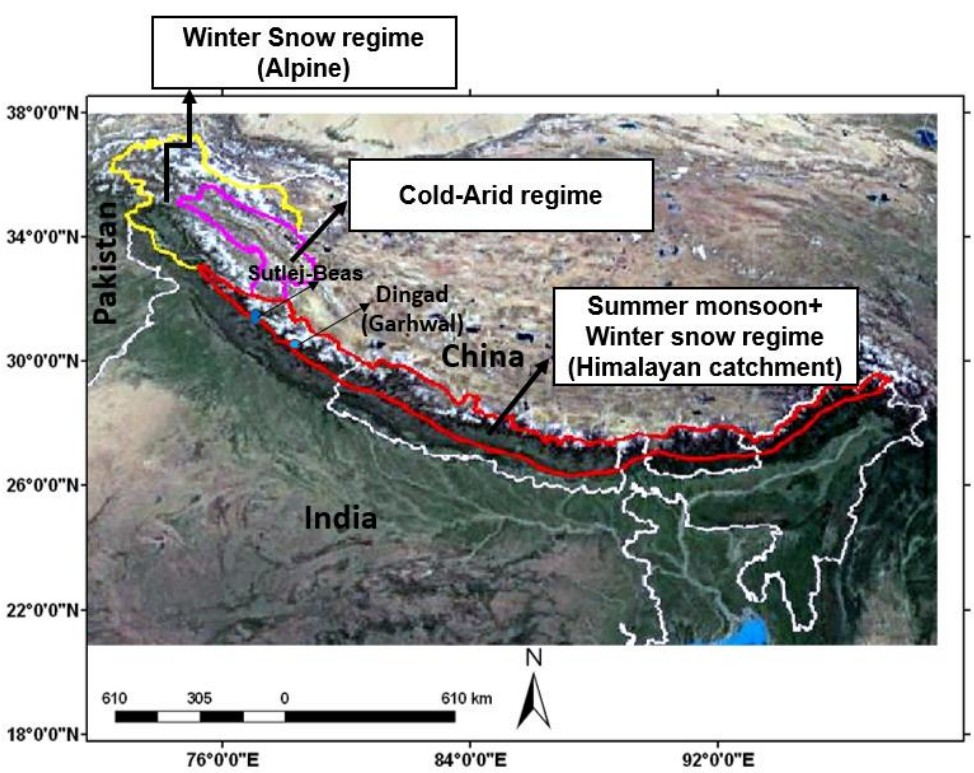

**Figure 1**. Glacio-hydrological regimes of southern slopes of the Himalaya and study area (After Thayyen and Gergan, 2010).



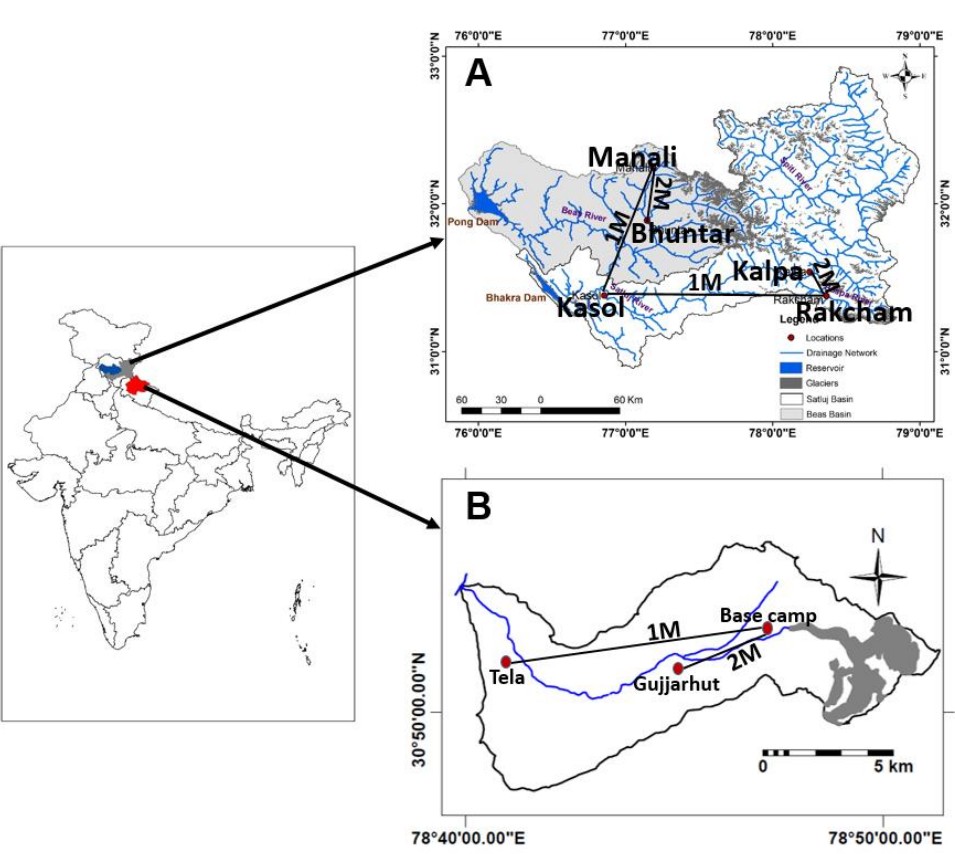

**Figure 2**. Study catchments in the monsoon regime: (A) Sutlej and Beas basin (B) Dingad catchment in the Upper Ganga basin.







**Figure 3**. Monthly distribution of temperature and precipitation at different elevations of the study basins. Sutlej – Beas basin (A,B,C) and Upper Ganga basin (D, E)





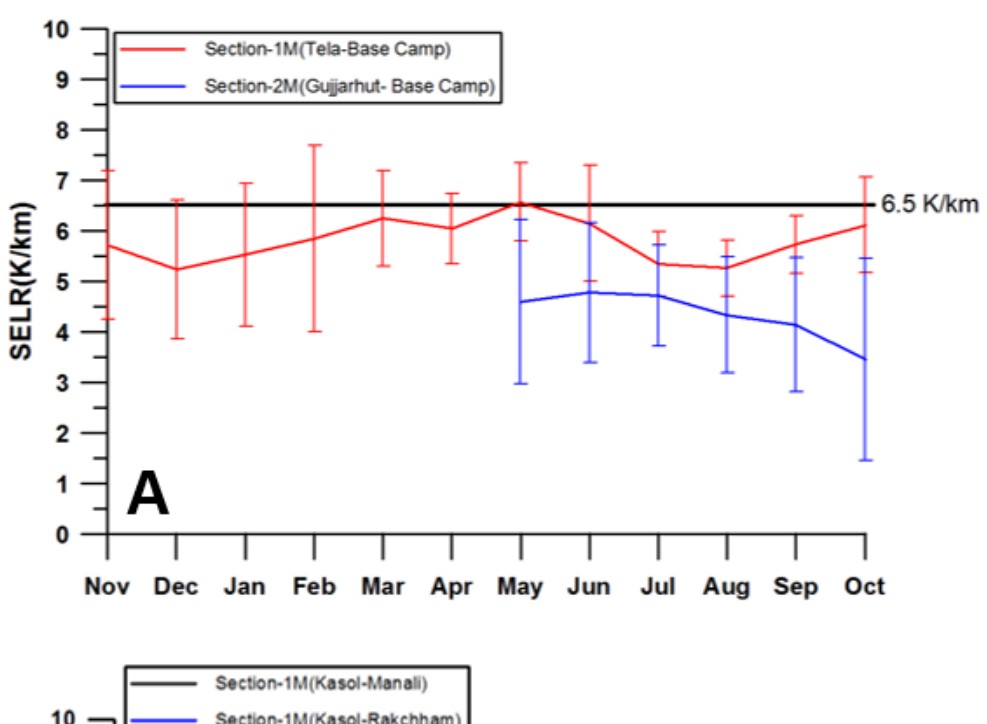

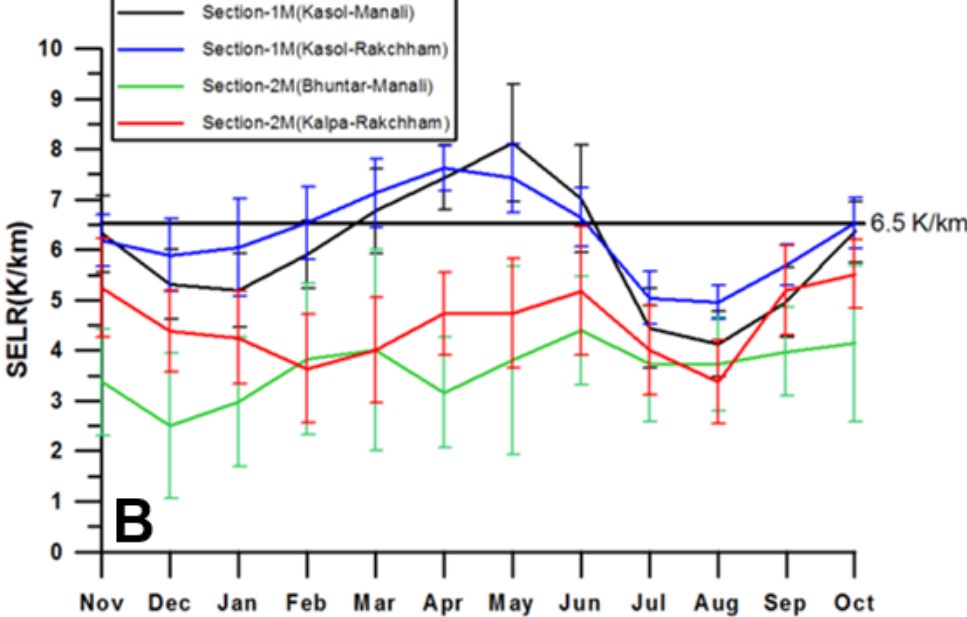

**Figure 4**. Monthly SELR variations of Section -1M and Section- 2M in the monsoon regime (A) Dingad catchment, Upper Ganga basin and B) Sultej and Beas basins.



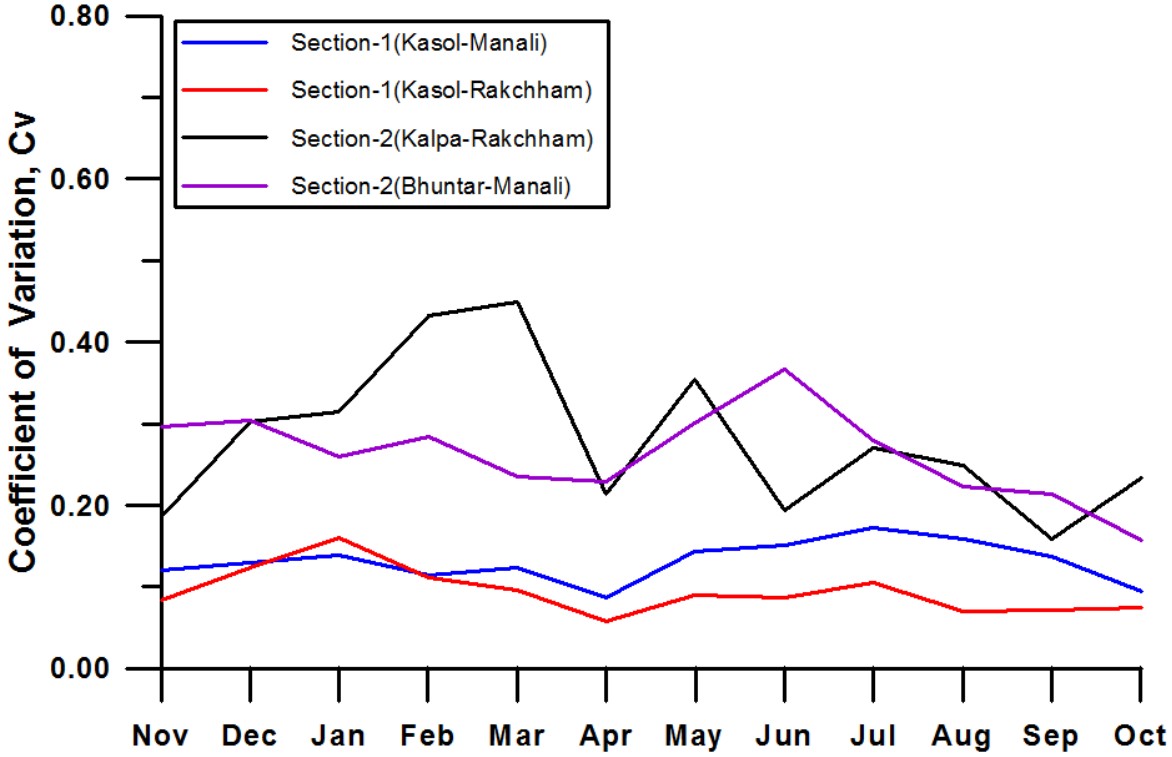

**Figure 5.** Coefficient of variation (Cv) of section -1M and Section-2M of Sutlej and Beas basins showing higher inter-annual stability of valley scale SELR( Section-1M) as compared to Nival- glacier regime SELR (Section-2M).





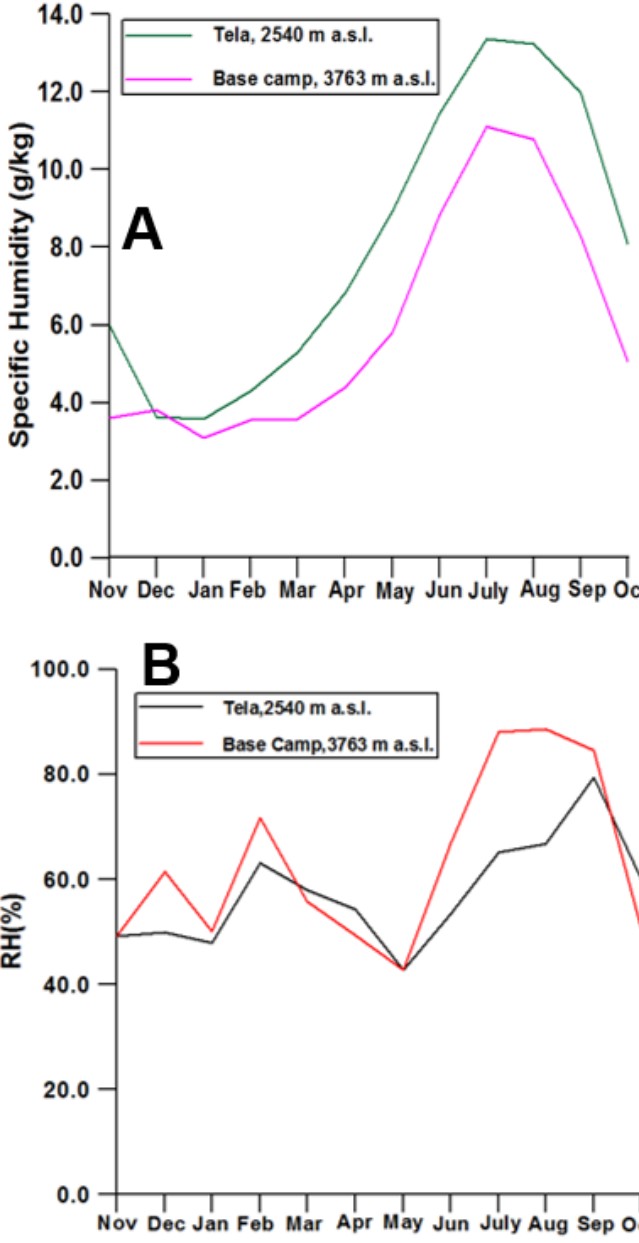

**Figure 6.** Mean monthly specific humidity variations of (A) lower (2540 m a.s.l.) and upper (3763 m a.s.l.) stations of Dingad catchment and (B) corresponding seasonal variations in relative humidity.



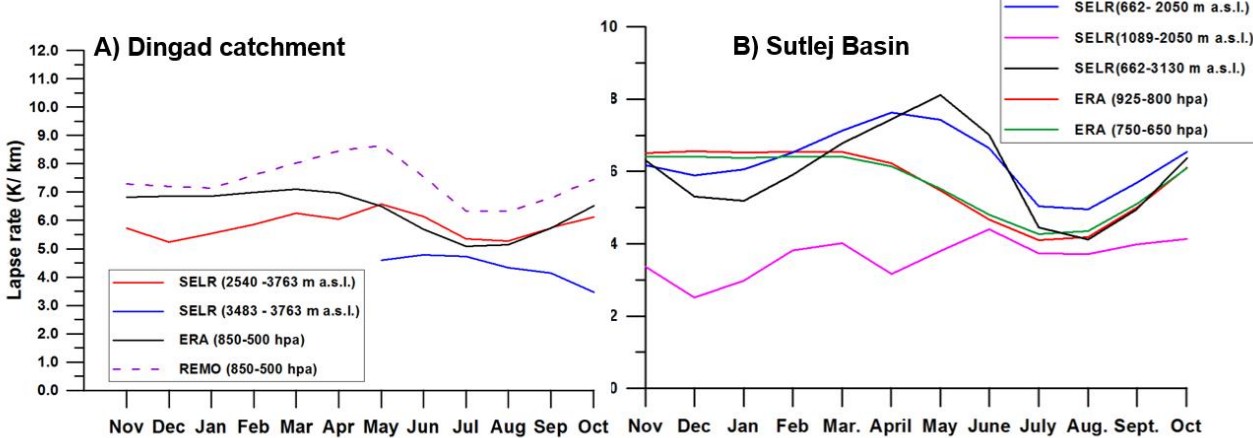

**Figure 7**. Comparison of ERA- interim free environment lapse rate with SELR of (A) Dingad catchment and (B) Sutlej-Beas basin. Free Environment lapse rate derived from REMO RCM for Dingad basin is also shown.

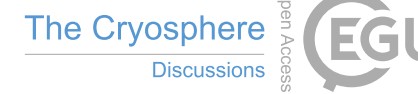

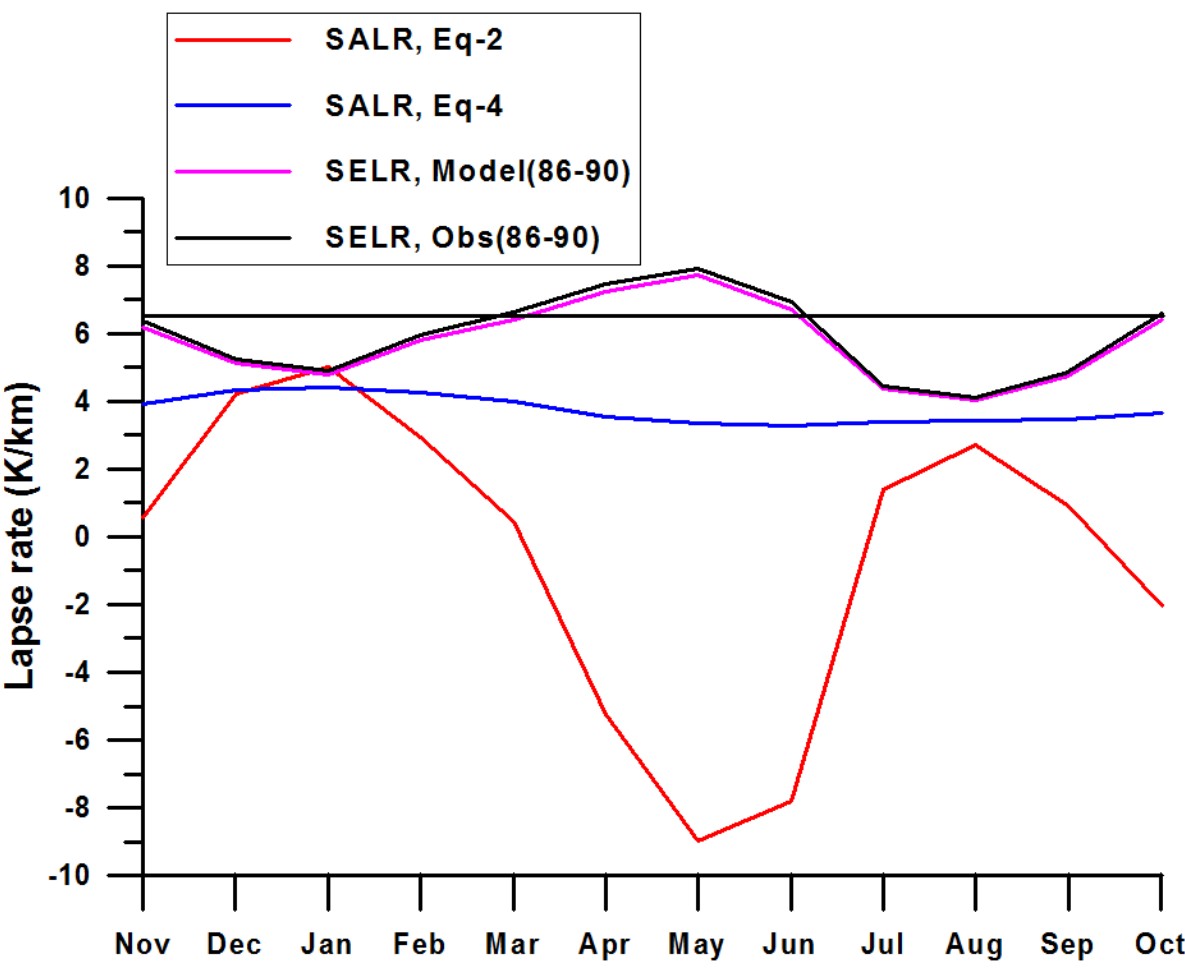

**Figure 8.** Temperature lapse rate of section- 1M of Beas basin between (Kasol, 662 m a.s.l. and Manali, 2050 m a.s.l.) calculated by using the Eq. 2 and Eq. 4 showing significant deviation from the observed SELR highlighting the need for a new approach. Modelled SELR of the section using derived indices shows promising result. Monthly averages of 5 year (1986-1990) were used here.




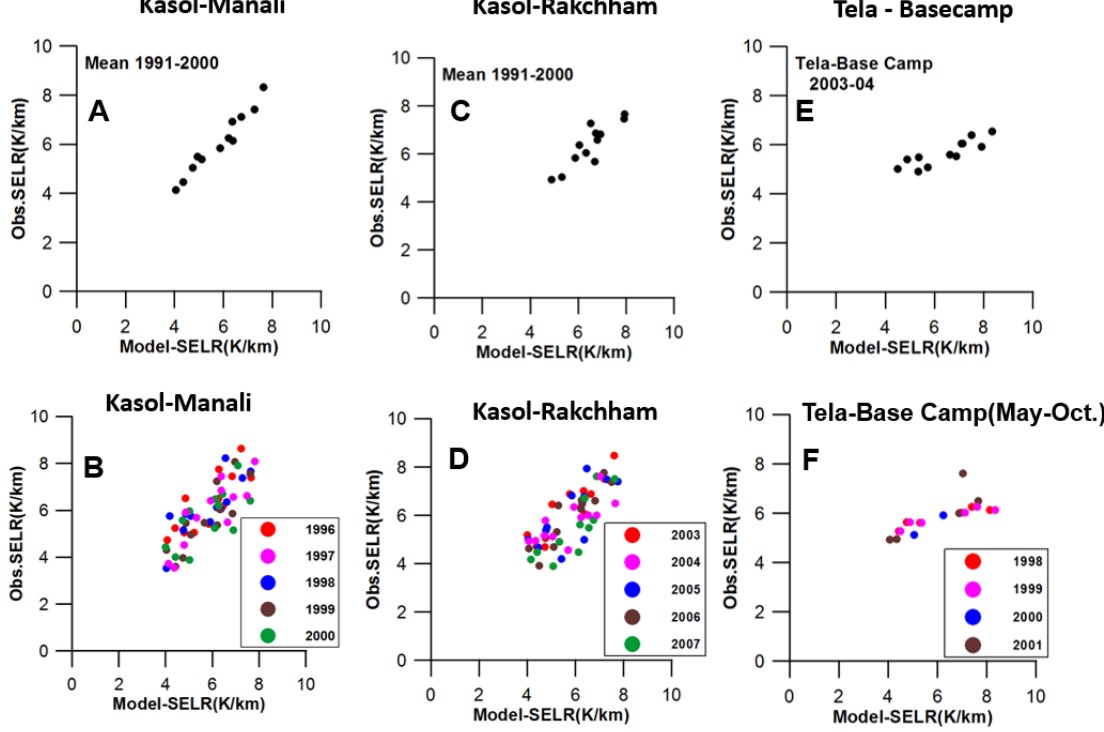

**Figure 9**. Relationship between modeled and observed SELR using monthly indices derived for Kasol-Manali section for 1986-1990 period. Derived indices provided satisfactory result for mean SELR of 1991-2000 period for Beas and Sutlej basins and Dingad catchment (2003-04) (A, C and E) as well as individual years (B,D and F).



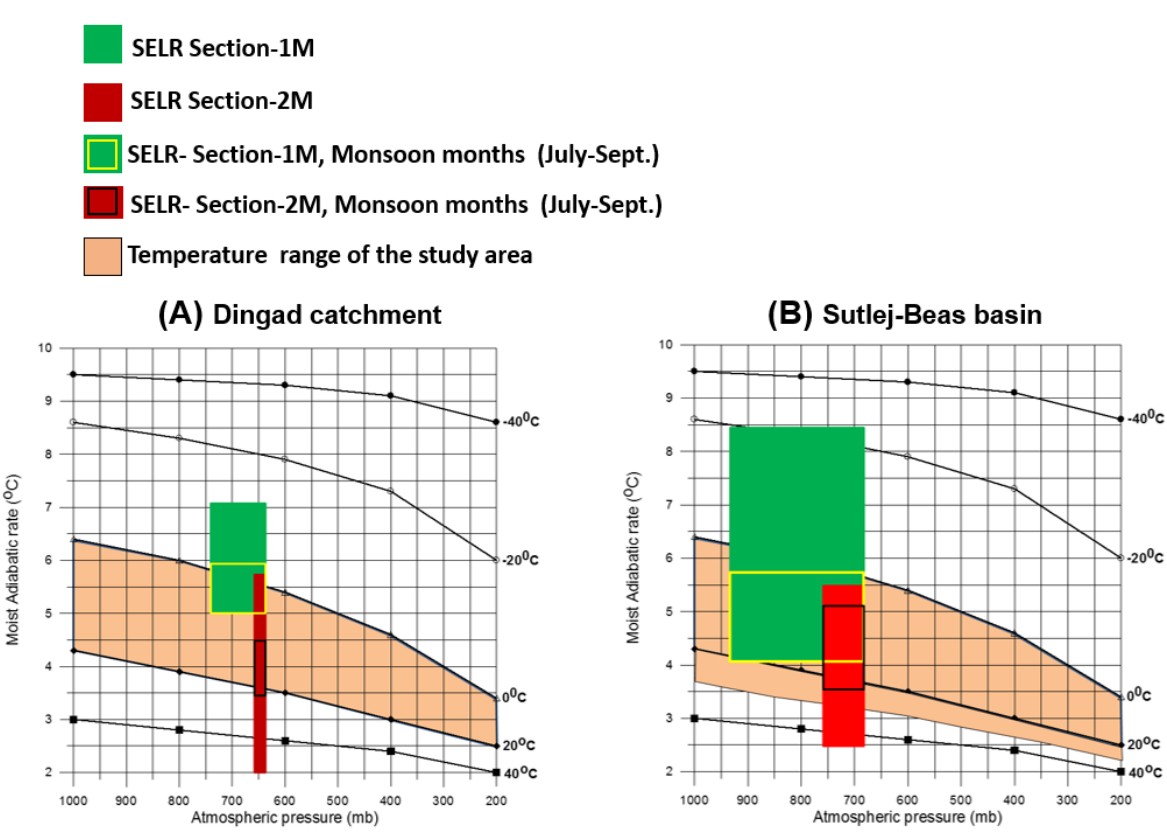

**Figure 10.** Theoretical saturated adiabatic lapse rate (SALR) under different pressure –temperature combinations calculated using Eq.4 and observed monthly SELR under monsoon regime (A) Dingad catchment and (B) Sutlej- Beas basin. Green blocks showing the extent of mean monthly SELR of Section-1M and Red squares for Section-2M. Yellow outline represent the SELR range of Section-1M during the monsoon months and blue outlines for section-2M. Both falls within the theoretical SALR range suggesting the influence of moisture influx and related monsoon lowering of SELR. Higher SELR of April, May and June months is also common to all the basins.



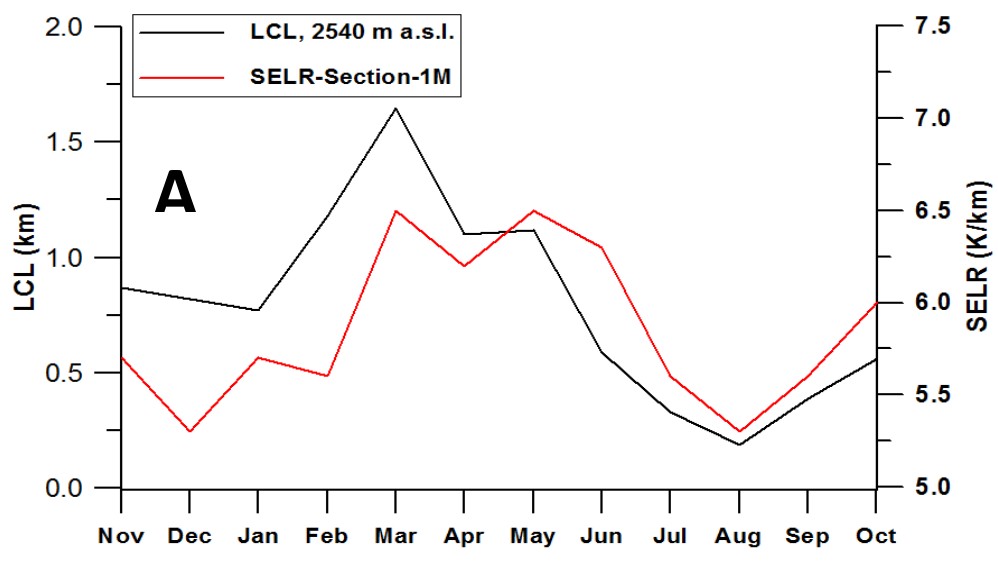

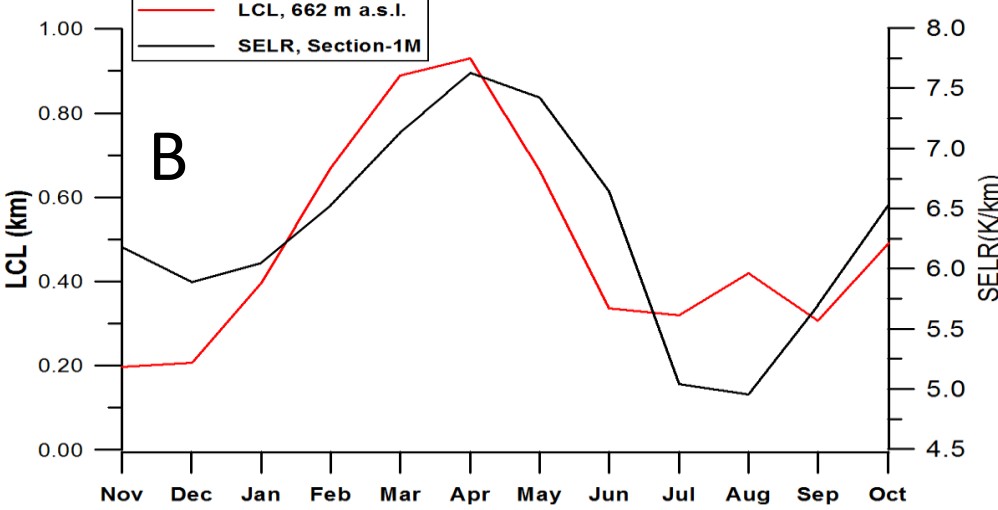

**Figure 11**. Seasonal variations of valley scale SELR is closely linked to the LCL variations in the monsoon region. Both A) Upper Ganga basin and B) Sutlej/Beas basin show similar response suggesting it as a regional characteristic influencing the valley scale SELR.