# Peer review of "Modeling Slope Environmental Lapse Rate (SELR) of temperature in the monsoon glacio-hydrological regime of the Himalaya"

_The Cryosphere, 2016_

## Short Comment (SC1) · 1 Sep 2016

In this paper Thayyen and Ashok are evaluating the monthly temperature lapse rate variations in the monsoon regime of the Himalaya.

I suggest to refer a key recent paper investigating the climate, its elevation-dependences, and temporal trends at high elevation in the Himalayan range.

Salerno F., N. Guyennon, S. Thakuri, G. Viviano, E. Romano, E. Vuillermoz, P. Cristofanelli, P. Stocchi, G. Agrillo, Y. Ma, and G. Tartari, 2015. Weak precipitation, warm winters and springs impact glaciers of south slopes of Mt. Everest (central Himalaya) in the last 2 decades (1994–2013). The Cryosphere 9, 1229-1247.

[Figure]

Pag.2 line 15-16 The authors show an increase in temperature and decrease in precipitation for the last twenty years using land meteorological stations.

From Pag.2 line 30 to Pag.3 line 7 Salerno et al., 2015 present the highest altitudinal gradient of the world (77–8848 m. a.s.l.). They found an altitudinal gradient of 0.60 °C (100 m)-1 on the annual scale with a linear trend. Furthermore they calculated the seasonal gradients and found a dry lapse rate of -0.65 °C (100 m)-1 during the pre-monsoon season when weather station registers a mean relative humidity of 62%. A saturated lapse rate during the monsoon season is -0.57 °C (100 m)-1 with a mean relative humidity of 96%. During the post-monsoon period, they found a lapse rate equal to that registered during the monsoon: -0.57 (100 m)-1 even if the relative humidity is decidedly lower in these months (44%). Kattel et al.. (2013) explain this anomalous low post-monsoon lapse rate as the effect of strong radiative cooling in winter.

---

## Editor Comment (EC1) · T. Bolch (Editor) · 12 Sep 2016

This is a revised submission of manuscript tc-2014-146 ("Factors controlling Slope Environmental Lapse Rate (SELR) of temperature in the monsoon and cold-arid glacio-hydrological regimes of the Himalaya"

http://www.the-cryosphere-discuss.net/tc-2014-146/

---

## Referee Comment (RC1) · Anonymous Referee #1 · 19 Sep 2016

Overall Comments: This paper evaluates the slope environmental lapse rate (SELR) variations in the monsoon regime of the western Himalaya, and suggest a modelling solution for the valley scale SELR assessment. In the Himalayan mountain system, the variations on SELR are mostly controlled by the moisture and orography, and with a significant seasonal variations, is a relatively basic findings of this paper. The authors further suggest the use of standard environmental lapse rate (free-atmosphere) for temperature extrapolate in the region is not appropriate. Fundamentally, as increased atmospheric water vapor content or inversions, as well as changing of topography can seriously influence the assessment of air temperature as function of elevation in a particular location, and in a particular time. Therefore, this kind of site specific work from

the Himalayan region could be useful for the wider reader not only in the field of snow-glacier runoff modeling but also modelling in various field, such as hydrology, forestry, ecology, agriculture etc. The paper is descriptive, and well written, however, there are still some issues that need to be fixed before consideration it for the publication. The comments are appended below.

Specific comments: 1. I felt that the introduction section is lengthy that could be shortened. The description of earlier studies from the surrounding regions is fewer, that mostly covered the information on the farthest mountain systems, thus, explained knowledge gap in this manuscript is shown weaker. In addition to this, the cited literature in this section look very old. Some latest information is also available, for example, authors can check Yao et al. (2012) and some others recent literature; I hope those literature can be useful to get some ideas of glacier variations in the HTP regions rather referencing of too old report (e.g., IPCC 2007). Authors have briefly outlined of sections in the introduction part, as I believe it is too conservative, and just kills the space. I also noticed some statements that included here could be shifted into the data and study sites section. 2. I am confused with the explanation of one major synoptic system that mentioned by the authors, such as "Indian Winter Monsoon (IWM)" that is responsible to carry moisture during winter season in the region. As per my knowledge, the monsoon in summer that originates in the Bay of Bengal, which is commonly known as the Indian Summer Monsoon or summer monsoon that affects the entire southern foot hills of the Himalayas, as well as the southeastern Tibetan Plateau on the northern sides. Generally, southern Himalayan region, particularly western parts with Nepal, north-western part of India, and northern mountainous region of Pakistan mainly influenced by the westerly synoptic system that brings moisture from the Mediterranean or Caspian Sea. How can it define as IWM, please make clear. 3. SELR variation is related to variations in topographic characteristics and pooling of cold air in the low-lying areas. Generally, the differences between the lapse rate for maximum and minimum temperature are likely to inversions for the minimum temperatures in the absence of wind and clouds. The inversion is the opposite effect of temperature lapse rate. This

[Figure]

phenomenon frequently occurs to the valley, and influences the mean temperature lapse rates. Your study sites are valley; thus, further investigation into this issue is suggested in the revised version. In addition to this, the variation in snow cover at higher elevation also influences the SELR. In your study region some sites are located at high elevation land, therefore, further assessment for the variation in SELR in relating to the variation on snow covers is suggested. For your reference, recent literature of Kattel et al. (2015) from the surrounding region could be useful. 4. Some arguments that included in the results section could be shifted into the discussion part; please check it carefully, and includes the results only that you have obtained from your own data. The authors have confirmed by citing earlier literature that the variations in the influx of moisture into the 'Himalayas catchments', its orographic lifting and resultant latent heat release during condensation are the major contributors for the SELR variations, however, still no clear discussion have made by the authors based on their own assessments because they have used numerous methods of hydro-statistic, as well as atmospheric thermodynamic system to interpret their results. Please also discuss clearly the effect of net radiation, orography, and especially the effect of turbulent heat flux in the discussion section before made the conclusions. One more missing explanation is that on page 16, in section 4.3 SELR and specific/relative humidity relationship; in this section the authors have just described the variation of specific/relative humidity, but no discussion is there of the relationship to SELR.

Minor comments: 1. On page 11, in para 5, Please define the DALR, and SALR 2. Somewhere the authors have used liquid condensation level (LCL) and somewhere the lifting condensation level (LCL), what is the difference? 3. Referring to Table 4; I have seen that there is a systematic differences of R2 and RMSE with increasing elevation. Please also explain the causes of variation of R2 and RMSE, accordingly in the text. 4. On page 26, Annexure (a and c), the value of a = 7.5, b= 237.3 is ok if T> 273.15, this coefficient is not valid if T ïĆč 273.15, did you check it? 5. Please merge the Figure 1 into Figure 2, and make only one. 6. There are still numerous typographical, grammatical and syntax errors throughout the text. I suggest to the authors please go

through it carefully and fixed all issues in the revised version.

---

## Author Comment (AC1) · 23 Sep 2016

These are responses to the major specific comments by the Reviewer-1 for the purpose of ongoing interactive discussion. A detailed reply including other minor points will be provided during the final response phase.

Response#1: Temperature lapse rates studies are very few in the Himalayan region, especially covering the higher altitude region , which reflects in the paper as well. The information from farthest mountain systems becomes all the more important because of the same reason. Those references provide an understanding of the advances made in the other mountain systems as compared to the Himalayan region. We believe, we have not missed any related critical references from the region and the references

suggested by the reviewer will be consulted in the revised MS.

Response#2: Latest research have provided better understanding of the wintertime precipitation mechanism in the Himalayas. Indian winter monsoon (IWM: during Dec, Jan and Feb) does carry the moisture from the mid Atlantic Ocean and Caspian Sea during the passage of Western disturbances (Dimri et al. 2015) embedded within the large scale sub-tropical westerly jet (SWJ). But moisture incursion is not limited from these two sources; moisture incursion from the Arabian Sea and Bay of Bengal as well added to the winter precipitation occurring over the Indian Himalayan fronts and thus this mechanism is called as IWM. Apart from Dimri and co-authors' work (not referred here but referred in the manuscript) many other researchers have given distinct dynamical and physical mechanisms of IWM, viz., Bony et al. (2000), Krishnamurti et al. (1997), Laat et al. (2002) etc. May be addition of latest work of Dimri (2016) will provide improved understanding on IWM. There are other work from synoptic analysis to diagnostics and modeling to provide dynamical and physical explanations on IWM. A discussion on IWM is out of scope of the present paper. Hence, some relevant references are provided here.

References

Bony S., W. D. Collins and D. W. Fillmore (2000). Indian Ocean Low clouds during the winter monsoon. J. Climate, 13, 2028-2043.

Krishnamurti T. N., B. Jha, P. J. Rasch and V. Ramanathan (1997). A high resolution global reanalysis highlighting winter monsoon. Part I, Reanalysis field, Meteorol. Atmos. Phys., 64, 123-150.

Laat A. T. J. and J. Lelieveld (2002). Interannual variability of the Indian winter monsoon circulation and consequences for pollution levels. J. Geophys. Res., 107, D24, 4739, doi: 10.1029/2001JD001483.

Dimri A. P. (2016). Warm pool/cold toungue Elnino and Indian winter monsoon. Mete-

orol. Atmos. Phys., 1 -11.

Dimri et al. (2015). Western Disturbances : A review. Rev. Geophys., doi: 10.1002/2014RG000460.

Dimri A. P. and A. Chevuturi (2016). Western Disturbances: An Indian Meteorological Perspective. Springer, 131pp.

Response#3: At vertical pressure/altitude levels maximum/minimum temperature hardly will occur at the same time. Thus the notion of the maximum/minimum temperature lapse rate at the same time is a debatable question.

Say, in case of the dry adiabatic lapse rate [dT/dz=-g/Cp] and saturated adiabatic lapse rate [dT/dz= -g/Cp-{(L/Cp)X(dws/dz)}] in principal, in situ maximum/minimum temperature observations will not satisfy the equation, as Cp will change at differing time of occurrence of maximum/minimum temperature at different altitudes. Use of daily mean temperature resolve this problem as it is not time specific. Diurnal micro to meso scale processes due to valley – ridge slope, higher elevation snow, inversion etc. can only be captured in the daily mean SELR rather than maximum/minimum temperature lapse rate. Therefore in the present manuscript SELR is specifically proposed and discussed as it very well captures net effect of suggested valley scale processes.(Descriptions of these equations are provided in the manuscript). Moreover, mean daily temperature is also the fundamental temperature unit used in many of the glaciological, hydrological and ecological models and our aim is to provide a better solution for improving such modelling efforts.

Response#4: What is suggested earlier is the lower lapse rate during the monsoon period. We have made significant contribution to understand the processes further from our own work such as a) differing valley scale (section-1) and higher altitude region (section- 2) SELR, b) Differing SELR stability of both the sections, c) Relationship with lifting condensation level variations d) Higher SELR for pre-monsoon season, e) SELR equivalence to SALR during the monsoon regime f) regional similarity in SELR for

both valley scale and higher altitude sections and g) proposing a modelling solution for SELR of monsoon regime etc. Again, role of radiation and other fluxes in SELR is not within the scope of the present manuscript primarily due to lack of data in this region, especially from the high altitude region. We hope that the present paper and discussion will trigger more research in this direction and emerging questions and issues will be addressed subsequently.

―――――――――――――――

---

## Referee Comment (RC2) · Anonymous Referee #2 · 17 Nov 2016

In this study, the authors analyse the Slope Environmental Lapse Rate (SELR) between several stations in the western Himalayas, and propose a way to deal with the annual cycle of the SELR by using an empirically tuned version of the Clausius-Clapeyron relation. While the objectives of the study are broad and well justified (who wouldn't like to have better temperature estimates for use in impact models?), the presented contribution has too many issues and general applicability problems to justify a publication in TC. One of the major messages of the study (abstract: "*Study suggests moisture-temperature interplay is forcing the seasonal as well as elevation depended variability of SELR*") is very basic and has been the core message of many previous studies with more stations and higher statistical significance (e.g. Kattel et al 2013 for Nepal,

mentioned in the manuscript). Beside being not mature enough (see discussion be-low), the modelling aspect of the study has a fundamental shortcoming which is only vaguely discussed by the authors.

Let me pick up some parts of the manuscript to make my point. In the manuscript (P9 L26), they write: "*SELR of section-2M representing the nival- glacier regime is considered important for cryospheric system studies in the Himalaya.*". Later (P12, L27), they state "*SELR modeling is attempted only for valley scale lapse rate (Section-1M) as lapse rate of nival-glacier system (Section-2M) is found to have higher inter-annual variability as discussed in section 4.2 above.*" As a glacier modeller (which I am), I am left with very confusing and contradictory messages here. The glacier lapse rate is important, but since it is complicated I shouldn't attempt to model it? To make things even more confusing, there is another statement in the abstract: "*Inter-annual variations in SELR of the nival- glacier regime is found to be significant while that of the valley scale SELR is more stable. Hence, it is proposed to use the valley scale SELR for glacier melt/runoff studies.*" This statement is not backed-up by any evidence, since the authors didn't try to apply their valley model to locations where glaciers and snow *are* found.

I encourage the authors to use the very valuable data presented here to reach higher goals, such as actual application of their method to modelling studies of the glacier melt/runoff. I also encourage them to be much more careful in proof-reading their manuscript before submission next time. There are few things able to annoy a reviewer more than incomplete/inexact figure legends, non-explained variables and acronyms, or typos in the abstract.

**Factors influencing the SELR**

In the manuscript, the authors explain the difference between the SELR and the free-atmosphere lapse-rate. The correction method they propose to apply, however, is a purely moisture based thermodynamic approach which has in fact nothing to do with the differences between the SELR and and the free-atmosphere. It could have (since the saturation level might be reached by orographic lifting), but this dynamical argument is not quantitatively discussed by the authors. Other reasons why the SELR is different from the the free-atmosphere are:

- differences in incoming radiation

- differences in surface conditions (roughness, albedo), changing the surface energy balance

- dynamical effects (e.g. föhn)

- cold-pools (nighttime inversions)

- etc.

All these effects are playing a role at the monthly time scale too, and can explain (i) why the "valley gradients" and the "nival-glacier" gradients are fundamentally different, and (ii) why the purely thermodynamical approach has to be corrected manually with empirical indexes (Eq. 7, Fig. 8). In fact, I am quite confident that it would be possible to reach the same modelling result as shown in Fig. 9 with a purely statistical approach, since it's just about adding a correction for the annual cycle to the SELR. The comparatively poor results of the model in simulating inter-annual variability speaks for the necessity for a more complex model.

**Evaluation of the model performance**

The authors calibrate their model at one location and apply it to the others. This is already a good idea, but should be extended to a full cross-validation: with cross-validation, you make use of all available data for calibration and validation, and verify the general applicability of your model for future users. If your model only works in one direction, it will probably have issues elsewhere too.

Furthermore, the choice of using correlation as an indicator of model performance is a bad choice. This is best shown in figure 9, upper-right and lower-right plots, where the correlation is good but where the absolute values are way off (model says 8, obs say 6). I would a very different approach for validation, by comparing your model results RMSE with a base method, i.e.: how well does my model perform in comparison to the standard lapse-rate 6 or 6.5 K/Km? How well does it perform in comparison to the average lapse-rate observed at that location, or an average annual-cycle of the lapse-rate? When using the two latter "simple models", the performance of your model can be put into context.

**Editorial comments**

A non-exhaustive list of issues underlining that authors should take more care of their readers before submission:

- many typographical errors, even in the abstract

- SALR is explained in the abstract but not in the text, DALR is explained nowhere (I know it means "Dry", but still)

- the formulas are unnecessarily difficult to follow: please be consistent in your

notations (e.g. $\frac{dw_s}{dt}$ is written in several different ways across the chapter) and consider using latex, which is free for everyone to use

- Eq 7: there is no explanation as to what we have to apply the sum to.

- Table 1 and 2: the legends are wrong, as there are both sections presented in each table

- Figure 4: there is no indication as to what the error bars are supposed to represent.

- the text is unnecessary long and contains many repetitions

- As a result of above and (I find) a quite intransparent choice for naming things (i.e. why "Section-1M" and "2M"?), it took me two lectures to get a sparse overview of the data availability.

---

## Author Comment (AC2) · 30 Jan 2017

**Author response to the review comments on "Modeling Slope Environmental Lapse Rate (SELR) of temperature in the monsoon glacio-hydrological regime of the Himalaya"**

We thank the reviewers for the thoughtful comments. We have undertaken a detailed revision based on the suggestions. Updated the figure 11(now Fig.10) with the SELR indices to show that the close relationship between SELR-LCL and SELR indices. New regional indices were derived by 5-fold cross validation as suggested. The improvement achieved through the cross validation is also presented. Detailed response to the review comments are follows.

**# Review-1**

1. Comment: Introduction section could be shortened

Response: Introduction has been shortened by following the reviewer suggestions

2. Comment: The description of earlier studies from surrounding region is fewer.

Response: Temperature lapse rates studies are very few in the Himalayan region, especially covering the higher altitude region. All the available information is included in the paper. The information from farthest mountain systems becomes all the more important because of the same reason. Those references provide an understanding of the advances made in the other mountain systems as compared to the Himalayan region. We believe, we have not missed any related critical references from the region and the references suggested by the reviewer is included in the revised MS.

3. Comment: Reference to the IPCC (2007) comment is old

Response: Reference to the IPCC(2007) is made to highlight the advancement made in the understanding of the Himalayan glacier response through post 2007 research to the focus. Yao et al 2012 added to the list.

"The IPCC (2007) report indicated a higher rate of glacier melting in the Himalayan region than elsewhere. However, systematic studies during the recent past have shown that the glacier change in the Himalayan region is comparable with other mountain glacier systems of the world except that of Karakorum region (Zemp et al., 2009, Bolch et al., 2012, Yao et.al., 2012)".

4. Comment: I am confused with the explanation of IWM

Response: Latest research have provided better understanding of the wintertime precipitation mechanism in the Himalayas. Indian winter monsoon (IWM: during Dec, Jan and Feb) does carry the moisture from the mid Atlantic Ocean and Caspian Sea during the passage of Western disturbances (Dimri et al. 2015) embedded within the large scale sub-tropical westerly jet (SWJ). But moisture incursion is not limited from these two sources; moisture incursion from the Arabian Sea and Bay of Bengal as well added to the winter precipitation occurring over the Indian Himalayan fronts and thus this mechanism is called as IWM. Apart from Dimri and co-authors' work (not referred here but referred in the manuscript) many other researchers have given distinct dynamical and physical mechanisms of IWM, viz., Bony et al. (2000), Krishnamurti et al. (1997), Laat et al. (2002) etc. May be addition of latest work of Dimri (2016) will provide improved understanding on IWM. There are other work from synoptic analysis to diagnostics and modeling to provide dynamical and physical explanations on IWM.

A discussion on IWM is out of scope of the present paper. Hence, some relevant references are provided here.

References

Bony S., W. D. Collins and D. W. Fillmore (2000). Indian Ocean Low clouds during the winter monsoon. J. Climate, 13, 2028-2043.

Krishnamurti T. N., B. Jha, P. J. Rasch and V. Ramanathan (1997). A high resolution global reanalysis highlighting winter monsoon. Part I, Reanalysis field, Meteorol. Atmos. Phys., 64, 123-150.

Laat A. T. J. and J. Lelieveld (2002). Interannual variability of the Indian winter monsoon circulation and consequences for pollution levels. J. Geophys. Res., 107, D24, 4739, doi: 10.1029/2001JD001483.

Dimri A. P. (2016). Warm pool/cold toungue Elnino and Indian winter monsoon. MeteC2 TCD Interactive comment Printer-friendly version Discussion paper orol. Atmos. Phys., 1 -11.

Dimri et al. (2015). Western Disturbances : A review. Rev. Geophys., doi: 10.1002/2014RG000460.

Dimri A. P. and A. Chevuturi (2016). Western Disturbances: An Indian Meteorological Perspective. Springer, 131pp.

5. Comment on Lapse rate of maximum and minimum temperatures..

Response: At vertical pressure/altitude levels maximum/minimum temperature hardly will occur at the same time. Thus the notion of the maximum/minimum temperature lapse rate at the same time is a debatable question.

Say, in case of the dry adiabatic lapse rate [$dT/dz = -g/C_p$] and saturated adiabatic lapse rate [$dT/dz = -g/C_p - \{(L/C_p) \times (dw_s/dz)\}$] in principal, in situ maximum/minimum temperature observations will not satisfy the equation, as $C_p$ will change at differing time of occurrence of maximum/minimum temperature at different altitudes. Use of daily mean temperature resolve this problem as it is not time specific. Diurnal micro to meso scale processes due to valley – ridge slope, higher elevation snow, inversion etc. can only be captured in the daily mean SELR rather than maximum/minimum temperature lapse rate. Therefore in the present manuscript SELR is specifically proposed and discussed as it very well captures net effect of suggested valley scale processes.(Descriptions of these equations are provided in the manuscript). Moreover, mean daily temperature is also the fundamental temperature unit used in many of the glaciological, hydrological and ecological models and our aim is to provide a better solution for improving such modelling efforts.

6. Comment on "The authors have confirmed by citing earlier literature that the variations in the influx of moisture into the 'Himalayas catchments', its orographic lifting and resultant latent heat release during condensation are the major contributors for the SELR variations, however, still no clear discussion have made by the authors based on their own assessments because they have used numerous methods of hydro-statistic, as well as atmospheric thermodynamic system to interpret their results."

Response: What is suggested earlier is the lower lapse rate during the monsoon period. We have made significant contribution to understand the processes further from our own work such as a) differing valley scale (section-1) and higher altitude region (section- 2) SELR, b) Differing SELR stability of both the sections, c) Relationship with SELR and lifting condensation level variations d) Higher SELR for pre-monsoon season, e) SELR equivalence to SALR during the monsoon regime f) regional similarity in SELR for both valley scale and higher altitude sections and g) proposing a modelling solution for SELR of monsoon regime etc. Again, role of radiation and other fluxes in SELR is not

within the scope of the present manuscript primarily due to lack of data in this region, especially from the high altitude region. We hope that the present paper and discussion will trigger more research in this direction and emerging questions and issues will be addressed subsequently. Moreover, model results suggests that the local effects are not a significant factor determining the valley scale SELR.

*Minor comments*: 1. On page 11, in para 5, Please define the DALR, and SALR : Done

2: Somewhere the authors have used liquid condensation level (LCL) and somewhere the lifting condensation level (LCL), what is the difference? Ans: Changed to lifting condensation level which is more appropriate as we are discussing orographic lifting.

3. Referring to Table 4; I have seen that there is a systematic difference of $R^2$ and RMSE with increasing elevation. Please also explain the causes of variation of $R^2$ and RMSE, accordingly in the text. Ans: Incorporated cross validation and improved model results as suggested by Rev-2

4. Please merge the Figure 1 into Figure 2, and make only one. Ans: Yes, Figures are merged

**# Review-2**

1. Comment: One of the major messages of the study (abstract: "Study suggests moisture temperature interplay is forcing the seasonal as well as elevation depended variability of SELR") is very basic and has been the core message of many previous studies with more stations and higher statistical significance (e.g. Kattel et al 2013 for Nepal).

Response: In addition to what is already known, we show here that the SELR is also elevation depended with higher altitude region showing a very different lapse rate equivalent to SALR. This elevation dependent variability is suggested for the first time and sentence has been changed to "Study suggests moisture- temperature interplay is not only forcing the seasonal variation but also the elevation depended variability of SELR". Second, we are showing that the monsoon influence on SELR earlier suggested for Dingd (Thayyen et al 2005) and Nepal (Kattel et al., 2013) is extend further to the north-west covering Sutlej/Beas basins. The paper also discuss the processes driving the SELR variations and offer a modeling solution. Hence provide significant addition to the existing knowledge. The information on the reduced SELR of higher altitude region and its higher inter-annual variations is very important as the glacier study strategy being practised advocates weather monitoring near to the glacier and extrapolation by using the environmenatal lapse rate. Here our results imply that even measured SELR of glacier regime may not be good for modeling future glacier response until we build a full atmospheric model to capture all the processes. There is lot of work to be done, mainly developing desired data and information from the high altitude region before we could reach there. We believe this paper will act as a trigger to achieve this long term objective. It is evident from the reviewer comment itself that the need for a full atmospheric model is being felt after our effort to bring in the thermodynamic formulation in this paper. In the data sparse Himalayan region, great effort and time is required to build concepts and further aceptance through supporting research. This is evident from the fact that after the lead author proposed the monsoon lowering of SELR in 2005 (Thayyen et al., 2005),it was a long wait to see a supporting paper from the region (Kattel et al 2013). Moreover, we have stressed on the SELR-LCL-SELR indices relationship as a very important process driving the seasonal variations in the valley scale SELR.

2. Comment: Let me pick up some parts of the manuscript to make my point. In the manuscript (P9 L26), they write: "SELR of section-2M representing the nival- glacier regime is considered important for cryospheric system studies in the Himalaya.". Later (P12,L27), they state "SELR

modeling is attempted only for valley scale lapse rate (Section-1M) as lapse rate of nival-glacier system (Section-2M) is found to have higher inter-annual variability as discussed in section 4.2 above." As a glacier modeller (which I am), I am left with very confusing and contradictory messages here. The glacier lapserate is important, but since it is complicated I shouldn't attempt to model it? To makethings even more confusing, there is another statement in the abstract: "Inter-annual variations in SELR of the nival- glacier regime is found to be significant while that of thevalley scale SELR is more stable. Hence, it is proposed to use the valley scale SELR for glacier melt/runoff studies." This statement is not backed-up by any evidence, sincethe authors didn't try to apply their valley model to locations where glaciers and snoware found.

Response: We see no contradictions in our statement. We are showing that the Nival- glacier regime has a different lapse rate. That means, 1. It is not good to use the standard environmental Lapse rate to extrapolate the temperature measured within the nival- glacier regime as practised very often 2. The elevation of the base station is important in deciding which lapse rate one should be using. It is not good to apply the valley scale lapse rate for temperature measured in the Nival- glacier regime and 3. One should be aware of the higher inter-annual variability of SELR of Nival- glacier regime while modelling future glacier and runoff response. We fully appreciate the need for a full scale atmospheric model but we are not in a position today to achieve that objective. So we propose the use of a stable valley scale SELR for modelling until a better solution emerge. Most of the high altitude region did not have temperature measurements and present study give confidence in using the lower elevation station data by using the proposed modelling solution. Figure 3, 4 and 9 explains these from our data. The repeatability of the model across the study region and different time periods suggests that the valley scale SELR have negligible influence of "local effects" as compared to Nival-glacier regime. This is one of the key take away message of this paper. We are determined to reach higher goals and application of this model for glacier/snow runoff is initiated under a nationally coordinated project already and we intend to present that paper separately. However, we strongly feel that the first step is to get acceptance to the methodology presented.

3.  Comment: In the manuscript, the authors explain the difference between the SELR and the free atmosphere lapse-rate. The correction method they propose to apply, however, is a purely moisture based thermodynamic approach which has in fact nothing to do with the differences between the SELR and the free-atmosphere. It could have (since the saturation level might be reached by orographic lifting), but this dynamical argument is not quantitatively discussed by the authors. Other reasons why the SELR is different from the the free-atmosphere are……..

Response: As mentioned in the previous section, the core message of the paper is the moisture dependency of the SELR. Figure -9 makes it very clear. This is further linked with LCL variations. We have updated figure 10 in the revised MS by showing the relationship between Valley scale SELR – LCL and SELR indices to explain this orographic effect. We can see here that the SELR indices itself is the quantitative expression. That is also supported by the very close results achieved through the moisture based modelling effort.

4.  Comment: All these effects are playing a role at the monthly time scale too, and can explain (i)why the "valley gradients" and the "nival-glacier" gradients are fundamentally different,and (ii) why the purely thermodynamical approach has to be corrected manually withempirical indexes (Eq. 7, Fig. 8). In fact, I am quite confident that it would be possibleto reach the same modelling result as shown in Fig. 9 with a purely statistical approach,since it's just about adding a correction for the annual cycle to the SELR. The comparatively poor results of the model in simulating inter-annual variability speaks forthe necessity for a more complex model.

Response: Our data and analysis presented here suggest that "all these effects" mentioned have minor role in determining the SELR. That is why the Nival- glacier gradients equals SALR and valley scale SELR get closer to SALR during the monsoons. Rest of the variability is explained by LCL and re-evaporation processes. That is why it shows systematic and cross the regional response from upper Ganga basin to Beas/ Sutlej basin where cold pools, dynamical effects, incoming radiation etc. could

be very different. Please note that the indices derived only through dws along the mountain slope. Role of the local effect is limited and may be responsible for the higher inter-annual variability as discussed before. Yes, it is possible to work with statistical approach as well. But our aim also is to understand the physical processes by which SELR vary and how the same processes are regionally valid.

Please note that The indices are developed by using just 5 years of data of one section (Kasol- Manali) following the comments of the reviewer of the first version of this paper (Thayyen and Dimri,2014). The reviewer and later the handling editor asked us to show that the model is valid for different time periods and at different geographical settings within the monsoon regime without any further calibration. Hence we have followed that advise and demonstrated that the model perform well across the region and different time periods. In the revised Ms, model performance is further improved by full cross validation as suggested by the reviewer.

5.    Comment: The authors calibrate their model at one location and apply it to the others. This is already a good idea, but should be extended to a full cross-validation: with cross validation, you make use of all available data for calibration and validation, and verify the general applicability of your model for future users.

Response: A five- fold cross validation for both the sections of Beas and Sutlej basin were performed and basin mean monthly indices has been derived as the final product. Further, it is tested for all the test folds (5 reach in both sections) and two years of Kasol- Rakchham section  as well  for the upper Ganga basin data which showed significant  improvement in the model result. Discussion also modifies accordingly in the revised manuscript. A new Table is added for the cross validation statistics. Further, The derived basin scale model is tested with the standard environmental lapse rate as well as suggested. Please note that correlation is shown as the one of the test statistics along with RMSE and P- value. We greatly appreciate the suggestions of the reviewer in this regard.

6.    Comment: Text is unnecessary long  and contains many repetietions

Response: Present MS is a resubmission of an earlier manuscript which was reviewed by 04 revieweres. Each of the revieweres asked to explain certain aspects and we have excecuted the same. As it is a resubmission we have carry forwarded all the suggestions of previous reviewers also in the present version.  In the revised MS we have pruned/edited many sections  as per the present reviewer observations.

Editorial comments

• SALR is explained in the abstract but not in the text, DALR is explained nowhere
(I know it means "Dry", but still): *DALR is explained*

• Eq 7: there is no explanation as to what we have to apply the sum to.: *'n' is number of years in the present case, added to the text*

• Table 1 and 2: the legends are wrong, as there are both sections presented in each table: *Corrected*

• Figure 4: there is no indication as to what the error bars are supposed to represent.: *Added in the title that error bars correspond to standard deviations*
• the text is unnecessary long and contains many repetitions: *Significantly reduced the text  and removed repetitions.*
• As a result of above and (I find) a quite intransparent choice for naming things (i.e.
why "Section-1M" and "2M"?), it took me two lectures to get a sparse overview of
the data availability: *Changed to Section 1& 2*

Comment: Suggested to refer the paper;  Salerno F., N. Guyennon, S. Thakuri, G. Viviano, E. Romano, E. Vuillermoz, P. Cristofanelli, P. Stocchi, G. Agrillo, Y. Ma, and G. Tartari, 2015. Weak precipitation, warm winters and springs impact glaciers of south slopes of Mt. Everest (central Himalaya) in the last 2 decades (1994–2013). The Cryosphere 9, 1229-1247.

Response: Suggested paper will be referred in the revised MS.